# Adaptive artificial evolution of droplet protocells in a 3D-printed fluidic chemorobotic platform with configurable environments

Juan Manuel Parrilla-Gutierrez[1], Soichiro Tsuda[1], Jonathan Grizou[1], James Taylor [1], Alon Henson[1] & Leroy Cronin [1]

Evolution via natural selection is governed by the persistence and propagation of living things in an environment. The environment is important since it enabled life to emerge, and shapes evolution today. Although evolution has been widely studied in a variety of fields from biology to computer science, still little is known about the impact of environmental changes on an artificial chemical evolving system outside of computer simulations. Here we develop a fully automated 3D-printed chemorobotic fluidic system that is able to generate and select droplet protocells in real time while changing the surroundings where they undergo artificial evolution. The system is produced using rapid prototyping and explicitly introduces programmable environments as an experimental variable. Our results show that the environment not only acts as an active selector over the genotypes, but also enhances the capacity for individual genotypes to undergo adaptation in response to environmental pressures.

---

[1] WestCHEM, School of Chemistry, The University of Glasgow, University Avenue, Glasgow G12 8QQ, UK. Correspondence and requests for materials should be addressed to L.C. (email: Lee.Cronin@glasgow.ac.uk)

Oil and water emulsions have long been linked with the process of abiogenesis because they provide self-assembled compartmentalization[1–3] and their compositional information[4] can transmit traits between generations[5, 6] despite the absence of information polymers, such as DNA or RNA. These systems are interesting since, although they lack a highly evolved metabolism, the semipermeability of the interface layer can maintain the emulsion integrity[7, 8] similar to how the cellular metabolism maintains a living cell. Moreover, they are able to respond to environmental chemical stimulus[9]. Similar systems can sense and interact with each other[10] analogous to cell signalling, and can be optimised with genetic algorithms[11].

Abiogenesis processes are of high importance for their application to the development of artificial life. Bio-inspired systems are gaining traction over mechanical robots creating artificial life because they can easily overcome some of the main disadvatges[12]. Energy efficiency is one of the main drawbacks of traditional robots because they require information to be transformed and sent back and forth between several different layers of abstraction[13]. This problem is accentuated in mobile systems where energy efficiency and response latency between the interfaces are critical. Behaviour-based robots using subsumption or reactive architectures were created to address some of these problems[14] especially the solutions which base its foundation in the robots being situated and embodied[15, 16]. These two concepts are central aspects for all forms of life known, where energy efficiency is crucial for survivability. Evolution is a process embodied in its environment[17, 18] because it delegates the evaluation and selection operations into it, easing its computational cost.

The process of evolution in the natural world occurs by selection of the fittest individuals, which results in their long-term survival and propagation. In the context of artificial chemical evolution, a well-known experiment by Spiegelman has demonstrated that even purely biochemical RNA molecules can evolve in vitro when they undergo an iterative process of amplification and selection[19]. In this experiment, RNA was replicated by viral RNA polymerase, Qβ replicase, in a test tube and a fraction of replicated RNA was serially transferred to a new test tube containing fresh Qβ replicase. Based on this system, it was also shown that RNA molecules encoding the Qβ replicase can evolve when mixed with cell-free protein expression system and compartmentalised as water-in-oil emulsions[20]. However, it is important to note that this selection occurs at the phenotype level, whereby the interaction of the living entities with their surroundings determines life or death at the ecosystem level[21]. Although the presence of complex and rapidly changing environments is the norm in the natural world, the impact of these on the evolutionary process is rarely reported beyond computer simulations[22]. This is important since the heterogeneity given from the environment leads to a complex interplay between the living species and the environment they live in, resulting in natural selection[23].

Here we develop an approach that uses rapid prototyping tools to design and manufacture hardware[24] that embodies an evolutionary experiment[16] based on a genetic algorithm[25, 26] involving selection of the artificial genotypes[4]. This approach allows us to define the environment as an experimental variable, an aspect that tends to be disregarded in artificial evolutionary studies outside computer simulation[27]. Therefore, we are able to modify the environment and study its effect on individuals during evolutionary experiments, emulating how a population of entities responds to sudden environmental changes and demonstrating that evolutionary selection can occur on non-living very simple chemical entities[3]. Furthermore, we can observe how these changes act as a population filter—reshaping and biasing artificial 'natural' selection.

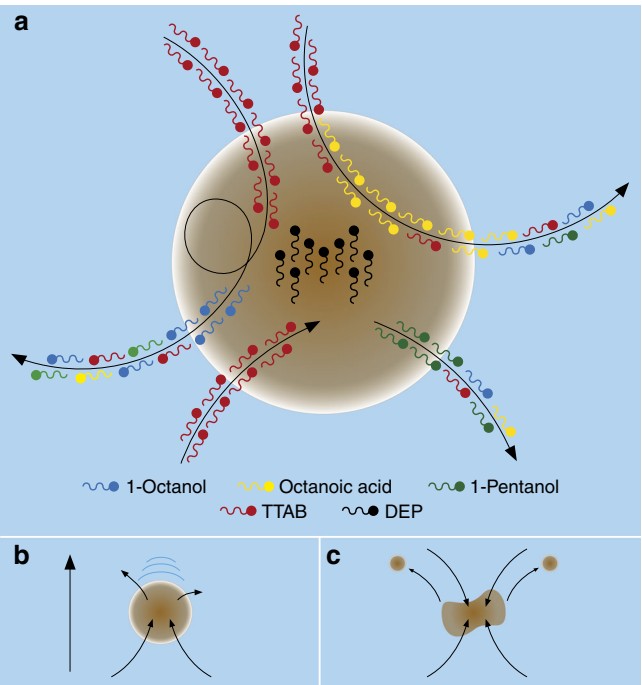

**Fig. 1** Droplets as seen from the surfactant perspective. **a** Our droplet system consisted of oil formulations of 1-octanol, 1-pentanol, octanoic acid and diethyl phthalate (DEP) in the oil phase, and tetradecyltrimethylammonium bromide (TTAB) at pH 13 in the aqueous phase. While the oil surfactants were non-ionic, TTAB is a cationic surfactant, meaning that the different surfactant molecules crossed the semi permeable interface until equilibrium. At the same time, the different oils dissolved at different rates into the aqueous phase following their respective solubility ratios. Based on the rate that these different gradients diminished, the droplets would **b** move or **c** divide

## Results

**Evolvable oil droplet system.** In order to explore the effect of the environment in an evolutionary experiment, we used simple oil-in-water droplets as model of protocells, which were formed by mixing four chemical components (1-octanol, diethyl phthalate (DEP), 1-pentanol and octanoic acid) and placing them into a water-filled environment (containing tetradecyltrimethylammonium bromide (TTAB) at pH 13), see "Methods" section. These compounds were selected to cover a wide range of different polarities, densities, viscosities, solubility and possible interactions that may occur at the interface (Fig. 1), with the purpose of creating droplets that would provide a chemical compartment related to the aforementioned attributes, aiming for their stability and the ability to move[28] and divide[29]. Previous work[11] demonstrated that even very simple oil droplet systems can display a range of complex behaviours essentially driven by the Maragoni effect[30] and furthermore that these behaviours can be altered by artificial evolution using a genetic algorithm (GA). The Marangoni effect stems from the partial dissolution of the components of the droplet into the aqueous phase. This changes the interfacial surface tension between the droplets and their surroundings. Due to small fluctuations in the surface of the aqueous phase, asymmetries in dissolution occur, and this results in unbalanced forces on the droplet, causing it to move. It is obvious that the solubility of the oil-in-water has a huge effect on this process, but other physical properties such as density and viscosity affect the movement of the droplets; for example, denser droplets sink below the surface while less dense ones spread out more on the surface. Less viscous droplets are

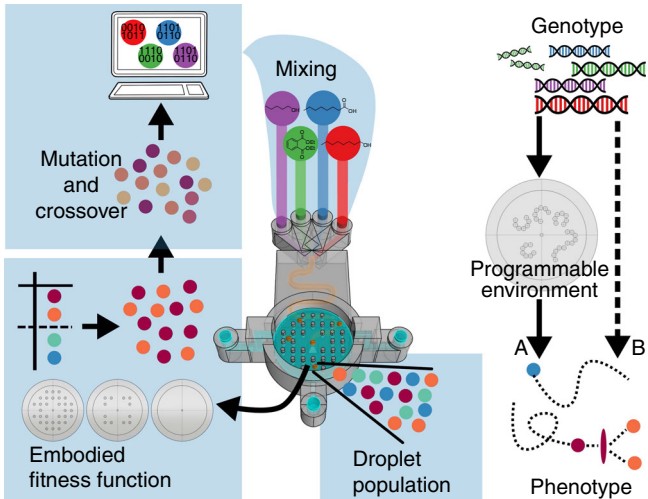

**Fig. 2** Schematic describing the evolutionary process. In the first step, a computer-generated random recipes using ratios of 1-octanol, octanoic acid, 1-pentanol and diethyl phthalate. These oils were then mixed through a serpentine channel, and populations of five droplets were generated in the evolutionary arena. The droplet behaviours were then analysed, and ranked using a fitness function. The best droplets were selected, and new droplet formulations were generated after "mutation" and "crossover" operations. This process continued through iterated generations. Because we used a 3D-printed device, we were able to change, during the course of the experiments, the physical environment in which the droplet population evolves. We thus explicitly studied how the genotype is modulated through a programmable environment to express its phenotype A, in contrast to the more studied genotype to phenotype direct approach B

more susceptible to deformation caused by their movement through the aqueous phase. Droplets also interact, as the positively charged surfactant head groups on the surface repel each other, preventing coalescence in most cases, while local pH gradients produced by the droplet also have an effect. Together, these physicochemical factors form a rich and complex system, which currently cannot be approximated with a physical model.

The droplets were represented by the formulation ratio of each oil in order to create a digital "genome". The behaviour of the droplets was characterised by image recognition and its results (e.g., number of droplets active at the end of the experiment) served as inputs to an evolutionary algorithm. The initially random formulations evolved during the following experiments in order to maximise the selected fitness function. By embodying evolution in a single device, we were able to extend its functionality not only as a test arena, but also as an additional parameter allowing for testing the droplet behaviours under different environments. During our experiments, the environment was defined as a set of obstacles which were monolithically 3D printed as a part of the device itself. The obstacles were made, just as the device itself, using polypropylene (PP), as it is chemically inert and does not react with the chemical compounds used. On the other hand, the obstacles physically played a role changing the behaviour of the droplets, for example, increasing the number of rebounds or reducing their area of movement. Thus, by studying how the environment changed the behaviour of the droplets at both the individual and population level, it was possible to study the evolutionary interplay between the compounds that acted as "droplet metabolism" and the obstacles that acted as artificial "ecosystem" (Fig. 2).

**3D-printed fluidic chemorobotic setup**. Our 3D-printed fluidic system was designed using CAD tools and printed using a commercially available 3D printer (see "Methods" section and

Supplementary Figs. 1 and 2). The objective was to encapsulate all the basic functionality from a liquid handling robot into a fully printable monolithic device. Throughout this study, the devices used did not show any apparent degradation after hundreds of experiments. PP is also one of the cheapest 3D printing filaments available, which coupled with the use of fused deposition modelling 3D printers, as fast prototyping tools allowed us to easily modify, iterate, prototype and manufacture new devices based on different requirements. Therefore, each characteristic of the design could be modified and a new device could be produced promptly within a short time at low cost, making our solution suitable for a large range of different experimental and laboratory requirements, compared to what was previously possible.

The main design had four inputs for the oil phases, which were mixed using a serpentine channel. In order to control the oil mixture, a series of liquid handling pumps were connected to the device and programmed to dispense the oils at different flowrates, depending on the ratio required by the experiment as defined by the digital genome. The serpentine channel led to an outlet situated at the bottom of the experimental arena, which generated the droplets and directly injected them into the aqueous phase (Supplementary Fig. 3). A typical experiment started by generating five identical 10-μl droplets within the experimental evolutionary arena. Once the droplets were placed into the aqueous phase, a video of the resulting droplet behaviour was recorded from above the arena using a camera (Supplementary Fig. 4). The arena also contained two additional inputs for the aqueous phase and cleaning solvent (acetone), as well as a drainage point used to wash away the old contents such that the system can be automatically reset for the next experiment (Fig. 3, Supplementary Figs. 5 and 6, Supplementary Movie 1 and Supplementary Note 1).

**Lattice search**. With the objective of fully exploring the formulation space, an evenly spaced combinatorial "lattice search" was conducted (see "Methods" section). The four different oils were taken into combinations of pairs, threes and fours, with a granularity of 10%, exploring in total 282 different oil formulations. Each formulation was studied five times within each of the three different environments. This exploration highlighted many unexpected behaviours: (a) the pillars would act as a droplet divider when moving droplets collided into them (Fig. 4); (b) the pillars would attract the droplets and inhibit their pattern of movement; (c) some droplet formulations would unhook themselves after a set time and then continue moving; (d) the pillars pulled the water meniscus creating valleys between them, where populations of low motility droplets locate; (e) the pillars could trap the droplets in a way that they lost their integrity and became dissolved (see Supplementary Fig. 7 for behaviours b–e).

**Evolutionary experiments**. After all the mechanical parts had been individually tested, the device was subjected to a series of evolutionary experiments in order to test its potential and check its statistical validity as an evolutionary platform. This was done using a fitness function that aimed to maximise the number of active droplets after 1 min of observation time, which is the same time scale we used in previous research[11] enabling us to validate this platform against it. More precisely, using the droplet detection algorithm described in the "Methods" section and Supplementary Fig. 4, we were able to obtain the number of droplets in a given frame by segmenting an image between foreground (droplets) and background. Our fitness function returned the number of detected droplets in the last frame of an experiment (considering experiments of 1 min and 30 frames per second, this meant the 1800th frame). As part of the detection algorithm, a droplet needed to move for about at least 3 s in order to be

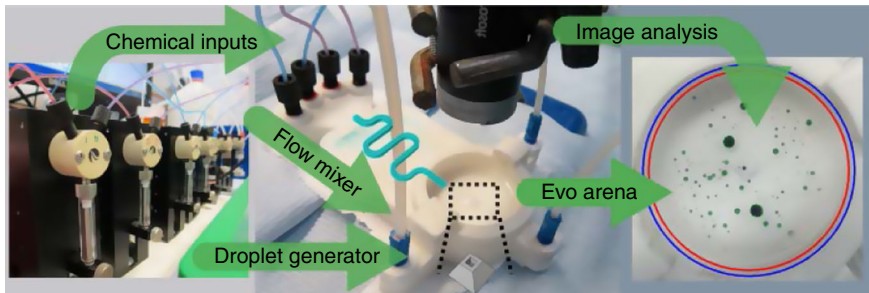

**Fig. 3** Schematic of the 3D-printed system. Chemical inputs: Seven liquid handling pumps were used for the oil phases, aqueous phase and cleaning processes. Droplet generator: The oil phases were mixed through a serpentine channel, and droplets were generated into an evolutionary arena. Evo arena: A camera recorded the arena from above. Image processing algorithms were used to analyse and categorise the droplets behaviours

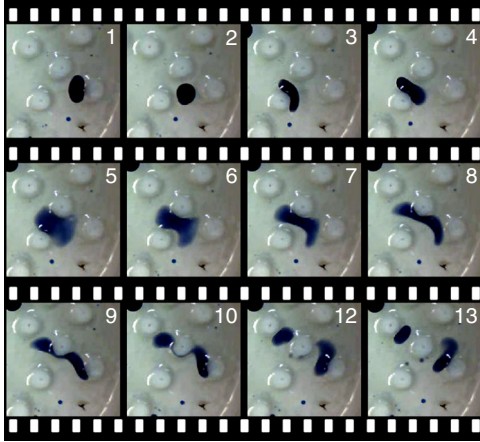

**Fig. 4** Droplet interactions with the obstacles. Time lapse images from 1 to 13. A droplet bounced against a pillar towards a second pillar that facilitated a split

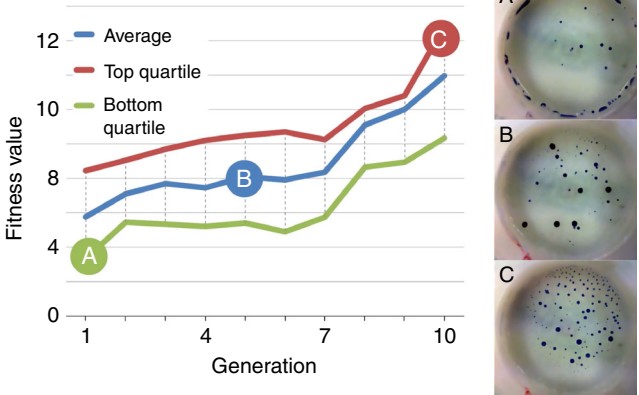

**Fig. 5** Fitness increase over successive generations and platform validation. Plot showing the change in fitness over each successive generation of experiments for the defined fitness function (droplet activity). The screenshots labelled A, B and C were taken using the last frame of the video experiments from the points labelled in the plot with the same letters

considered part of the foreground. Therefore, experiments where the droplets underwent controlled division and movement were assigned correspondingly high fitness values. At the end of the initial set of experiments, a genetic algorithm (GA) was used to evolve the droplet against this fitness function (see "Methods" section). The GA was run for 10 generations with 20 individual genomes per generation. For each genome, five different experiments were done and the average fitness used. At the start of the GA experiments, the genomes used in the first generation were selected randomly, and successive generations were built using fitness proportionate selection to choose 10 parent candidates, and adding 10 new offspring by stochastically choosing pairs of parents and performing a one-point crossover followed by a mutation operation. Figure 5 shows the result from a single GA run (see Supplementary Figs. 8–11 for more repetitions). The average, top quartile and bottom quartile values shown in Fig. 5 were calculated per generation. Each generation had 20 individuals, and each individual was repeated five times. The fitness value of an individual was the average of its five repetitions, and the average fitness value of a generation was the average fitness value of the 20 individuals, while the top quartile represented the top 25% individual (in our case, the fifth individual), and the bottom quartile represented the bottom 25% individual (in our case, the fifteenth individual). In this case, the number of droplets nearly doubled after 10 generations of droplet evolution, effectively maximising the defined behaviour. This result demonstrates that the 3D-printed device could perform evolutionary experiments on the defined droplet system, replicating the capabilities of a liquid handling robot[11].

Taking advantage of the modular and digital design of our device, the arena could be transformed into different environments in which the evolutionary experiments are conducted, thus adding a new degree of freedom to the system. In addition to the empty arena that defines the first environment, we developed two other environmental arenas: an arena consisting of pillars, and one generated using a Lidenmayer system[31]. The pillar-based arena is composed of a matrix of pillars, and was manually designed in order to be densely populated with obstacles providing a significant set of barriers to the droplets. This was to ensure some interaction between the arena environment and the protocell droplets undergoing the artificial evolution process. The other arena was algorithmically generated using an L-system, which is a digitally defined and parameterised environment-generating system (see "Methods" section). L-systems are widely used in biomimetic robotics to describe procedurally generated organisms and ecosystems[32]. Introducing stochastic decision rules in the procedure, we were able to generate a wide range of unique landscapes, and one of them was deliberately chosen based on its structural arrangement as a cave-like shape of obstacles, because we expected this type of arena to have a substantial impact on droplet behaviours. We repeated the GA with these two new arenas, and the fitness of the populations also increased through generations (Supplementary Figs. 12–15). This demonstrates that our device is suitable for evolving populations of droplets in heterogeneous environments.

In order to emulate how rapid environmental changes affect evolution, a GA run was performed where the initial 10 generations used the empty arena, the following 10 generations used the pillared arena and the last 10 generations used the

L-system arena, see Fig. 6 and Supplementary Fig. 16. After the first environmental change, the fitness of the droplets population dropped by nearly half, proving that our environmental modifications had an impact over the droplet's behaviours. In

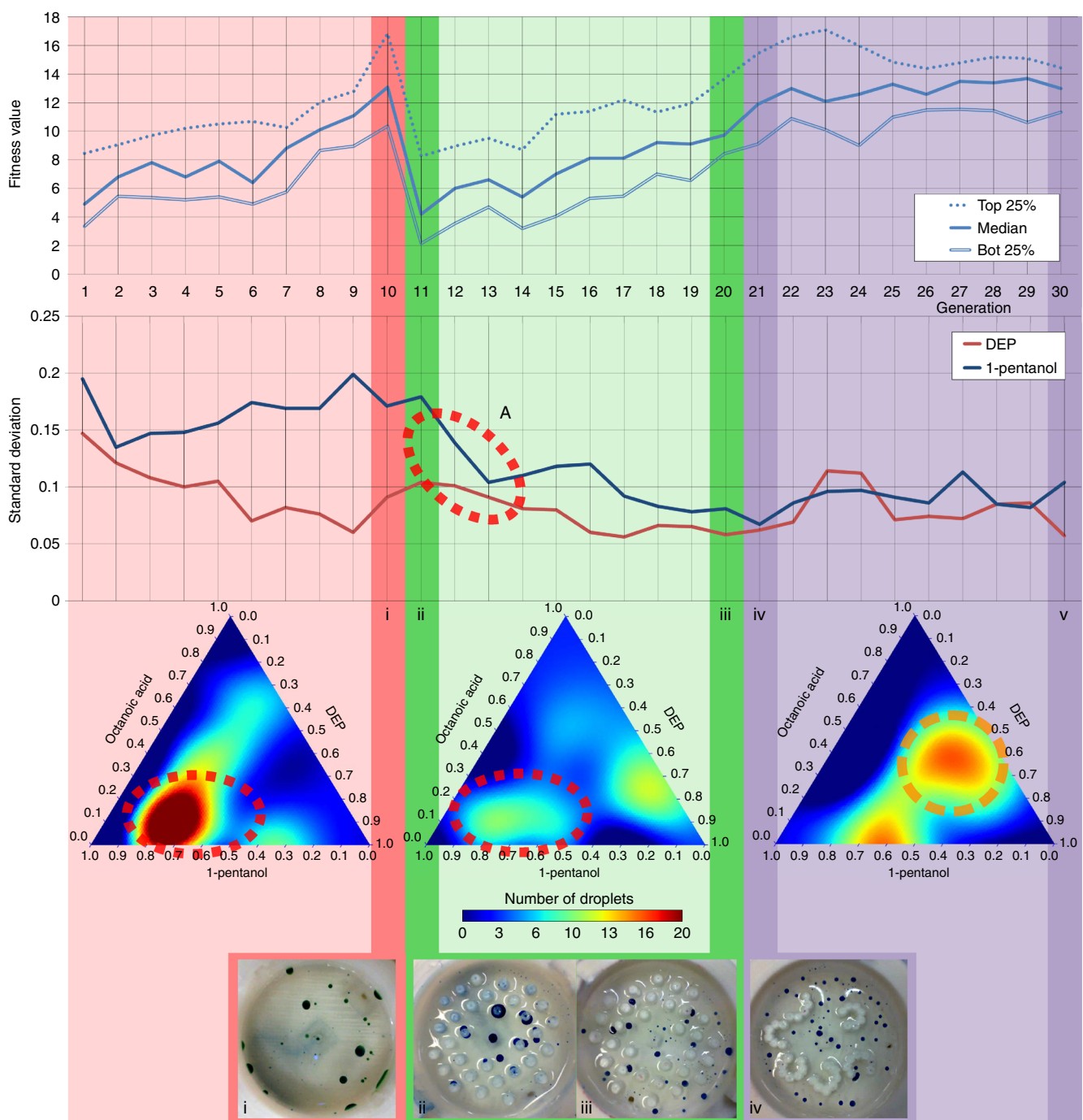

**Fig. 6** Environmental change model. Fitness value and standard deviation linear plots: Charts plotting the fitness statistical values and genome standard deviations (SD) of each generation through time. The first 10 generations used an empty arena, the following 10 generations used an arena filled with pillars, and the last 10 generations used the arena generated using an L-system. The arena was swapped to the pillars environment between the 10th and 11th generation, inducing a drop in the fitness values, which forced the droplet to adapt and evolve again in the new environment. The arena was swapped to the L-system between generations 20 and 21, introducing no noticeable fitness changes. Points of interests are: **A** where after the first swap of arena the genome SD drops, because the new environment acted as a "filter", choosing a small subset of the population as viable candidates. These results can be corroborated in the middle fitness landscape, where the initial one shrank into a small island. **B** where after the second swap of arena a new phenotypic island appeared where before the same genotypes were producing very low fitness values. Full fitness landscapes in Supplementary Fig. 24. Picture "**i**" represents an example of a recipe in the last generation of the empty environment. Picture "**ii**" represents the same recipe but now in the pillars environment. Picture "**iii**" represents an example of a recipe in the last generation of the pillars environment. Picture "**iv**" represents the same recipe but now in the pillars environment

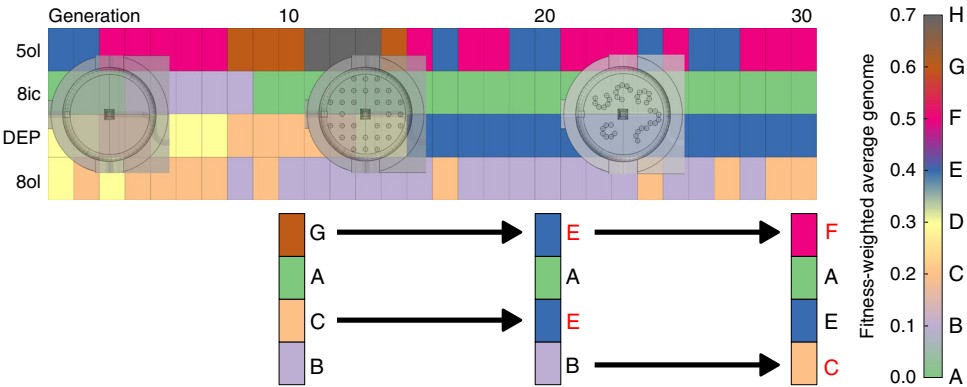

**Fig. 7** Heat map evolution of the average genome through the three environments. Each of the columns in the heat map represents a generation, and each of the rows represents one of the four components that formed our droplets. For each generation, its average genome was calculated, and each of the components was rounded to one decimal, discretising the genome space into 10 slots, which were assigned a different letter. Using this convention, it can be seen how the different environments introduced significant genome variation in two of the components for each new environment. "5ol" = 1-Pentanol, "8ic" = octanoic acid, "DEP" = diethyl phthalate and "8ol" = 1-octanol

fact, we observed a "filtering" effect due to the environmental change, as indicated by the fitness landscape models where the optimal genotypes of the pillars environment were a subset of the genotypes selected in the first environment (Fig. 6, bottom ternary plots, highlighted as "A"). In contrast, the second change of arena introduced a different effect on the evolution dynamics, because the third environment resulted in a fitness landscape where new suboptimal behaviours appeared (highlighted as "B" with an orange broken circle in Fig. 6), where no peaks were observed in the first environment, analogous to the concept of phenotypic plasticity[33]. See Supplementary Movies 2 and 3.

**Statistical analysis**. A genomic analysis showed that after 10 generations of our single GA run, the fitness-weighted average genome (WG) for each of the environments was (octanol%, dep%, octanoic%, pentanol%): ($15 \pm 11$, $23 \pm 7$, $13 \pm 11$, $49 \pm 14$, mean $\pm$ s.d.) for the empty arena, ($13 \pm 7$, $22 \pm 5$, $10 \pm 11$, $55 \pm 15$) for the pillars arena and ($17 \pm 9$, $30 \pm 12$, $8 \pm 10$, $44 \pm 11$) for the L-system arena. A one-way ANOVA test comparing the WG of each of the arenas gives statistical significant $p$ values for the DEP (0.01) and 1-pentanol (0.002) components.

The WG for the pillars environment after 20 generations using the empty environment during the first 10 generations was ($11 \pm 6$, $39 \pm 6$, $7 \pm 7$, $43 \pm 9$). An ANOVA test comparing this population, against a GA run, where only pillars were used shows significant $p$ values in DEP, octanoic acid and 1-pentanol ($1.25e{-}17$, 0.004, $1.99e{-}07$). Finally, the WG when the first 10 generations used the empty environment, the following 10 pillars environment and the last 10 that used the L-system environment (Fig. 6) was ($10 \pm 10$, $38 \pm 7$, $7 \pm 6$, $45 \pm 8$). An ANOVA test comparing this population (evolved on empty then pillars then L-system) against a GA where only the L-system environment was used shows significant $p$ values in the DEP (0.0004) and 1-octanol components (0.006).

This analysis shows that the evolution in each of the arenas leads to a differentiation of droplet population in each environment as indicated by their significant genome differences. In addition, and in every case, the final genome differed when environmental changes were introduced as opposed to when a single environmental was used all along. This is shown in Fig. 7 whereby a genetic colour "heat map" is shown and shows that the genome at the end of the empty arena run is "GACB", which then changes to "EAEB" after evolution in the pillars-array, to finally "FAEC", after evolution in the "caves" arena (Supplementary Figs. 17–19 for more information). This suggests that the

evolutionary pathway was influenced by the environmental history, effectively driving the evolutionary process into a different fitness niche than the immutable environment. It illustrates that the environment and its changes through time plays an active role in the development and expression of our droplet system.

To further study the impact of environmental modifications in our droplet system, we tested the last generation of droplets evolved in each environment (in Fig. 6, these generations were marked with "i", "iii" and "v") in the other environments (Supplementary Tables 1 and 2). Our results show that while the best individuals from the empty arena had reduced fitness when placed in the pillars or caves arena, the best individuals from both the pillars and caves arena kept a similar fitness when tested in the empty arena. This seems to indicate that the empty arena provided a "wider environmental niche", where droplets evolved to, but which were not able to survive in more constrained environments. On the other hand, both pillars and caves seem to have narrower environmental niches, requiring the droplets to evolve specific traits to survive, producing more robust recipes. Such recipes could perform well in the other environments because of a stronger environmental pressure for the traits that are important for "survival", resulting in a final subset of recipes able to perform in each arena. These results are similar to the biological concept of phenotypic plasticity, which is the ability of an organism to change its phenotype in response to changes in the environment. Our system was able to manifest this phenomena, but instead of using full-fledged biological entities, we used simple chemical ones, only formed by lipid molecules, thus reinforcing the idea that such simple entities could have had a role during abiogenesis.

## Discussion

By using a 3D-printed fluidic device with programmable environments, we were able to embody evolution in an experimental monolithic setup. This means that it was the interplay between chemistry and the generated physical obstacles that allowed us to explore the potential of oil droplets to evolve under a set of given conditions, and then adapt and continue evolving once these conditions were modified. We have shown that simple oil-in-water droplet formulations are a viable model for evolvable protocells even though they do not contain any kind of sequential information. Moreover, we showed that populations of droplets can overcome rapid environmental changes and continue their evolutionary process in the new arena, akin to how living entities

adapt to new habitats. Not only can our platform be used to optimise oil-in-water droplet formulations, but it can be easily extended by increasing the number of inputs, adding different reagents or evolving the physical experimental arena itself. This research also highlights that engineering the environment itself as a variable is a promising approach for the optimisation of complex systems[34]. Indeed, our platform is flexible and can act as a sandbox for probing chemical spaces for the discovery of new types of formulations or soft materials, whereby the environmental parameters are set as "engineering specifications". Our results show that by reshaping the physical environment, we were able to influence the evolutionary pathway through a dynamic modification of the fitness landscape, akin to natural evolution.

## Methods

**Experimental protocol.** Once the computer provided a recipe as described via a genetic algorithm, a lattice search or any other method, the 3D-printed device was programmed to prepare the mixture, execute five experiments and clean the arena in order to get ready for the next recipe. In order to mix the oils through the serpentine channel, the oil with the highest value in the recipe was set to a volume of 400 μl, while all the other oils were given a volume proportional to 400 μl based on the ratio defined in the recipe. These volumes were then pumped into the device and through the serpentine channel at a flowrate derived from the proportional ratios. The oil with the highest volume was pumped in at 2.5 μl per second, while the other oils were pumped in at a slower speed proportional to the recipe. These oil mixtures were dispensed into the device arena, and once the mixture was completed the device removed the contents into a waste drum, and the arena was washed with 2.5 ml of acetone. This process was not only used to prepare a new mixture, but also to remove the contents from the previous mixture. This mixing process was repeated three times. After this, a series of cleaning cycles were executed in order to be sure that the arena was fully cleaned. The first step was to wash it with 3 ml of acetone, and remove the contents. Then 2.5 ml of aqueous phase, and remove its content. Then 10 ml of acetone, and remove its content. This step was executed twice. Finally, 3.75 ml of aqueous phase, and remove its content. This step was executed twice. The next objective was to perform the experiment. In order to do so, the arena was initially filled with 3.75 ml of aqueous phase, and then five oil injections of 10 μl were executed to generate the oil droplets. At this point, the experiment was performed and it was recorded using a camera. Each experiment lasted 1 min. The next objective was to clean the arena in order to perform the next experiment. In order to do so, the arena was initially filled with 7.5 ml of acetone, and its contents removed. Then it was filled with 3.5 ml of aqueous phase, and its contents removed. This last step was repeated twice. Once this last step was performed, the device would fill the arena with aqueous phase to execute again the same recipe (each recipe was tested five times), or a new recipe would have been sent from the computer, in which case the device would start again with the mixing procedure.

**Preparation of solutions.** Initially, NaOH (20.0 g) was dissolved in distilled water (ca 4.8 l), then TTAB (33.65 g) was added. Finally, the pH was adjusted to pH 13 using 6 M NaOH solution, and the volume was adjusted to 5 l. The pH metre used was calibrated between pH 7 and 10. The oils 1-octanol, 1-pentanol and DEP were prepared in 200 ml aliquots in reagent bottles, while octanoic acid was diluted with 1-pentanol (20% octanoic acid 80% 1-pentanol), and also prepared in 200 ml aliquot in a reagent bottle. DEP and 1-octanol were dyed with 0.25 mg ml$^{-1}$ Sudan II blue and vortexed to mix. 1-Pentanol and octanoic acid were dyed with 0.25 mg ml$^{-1}$ Sudan III red and vortexed to mix.

**3D-printed device manufacturing.** The devices were designed using the CAD software "Rhinoceros 5". The 3D models were exported into STL files, and the STL files were transformed into "G-code" using the software "Bits from Bytes Axon 2", see Supplementary Fig. 1 for the configuration used. The devices were printed using the 3D printer "Bits from Bytes 3D touch". PP filament with 3 mm diameter was used. Both transparent and white filament were used without any significant difference. Supplementary Fig. 2 shows the base design of the device used during the experiments. Different variations of this device only added obstacles into the arena. Once a device was 3D printed, the only manual operation required was to tap its inlets in order to be connected to tube connectors. In order to do so, a thread of size M6 was used.

**3D-printed fluidic platform.** A fully automated fluidic device capable of producing droplets in a Petri dish-like arena with aqueous subphase was constructed. The device was a monolithic 3D-printed piece using a commercial 3D printer in its standard configuration. The device had a series of inputs and outlets connected to commercial liquid pumps. The pumps used were defective "Tricontinent C-Series Syringe Pumps". Their stepper motors were connected to "Pololu a4988" drivers. The drivers were controlled using an "Arduino Due" board. A homemade PCB

shield was used to easily connect the stepper drivers to the Arduino board (Supplementary Fig. 5), and custom firmware was written using the Arduino suite in order to control the pumps. The syringe pumps used 500 μl syringes for the four oil phases, and 5 ml syringes for all the other pumps. The pumps used three-way PEEK valves, as provided by Tricontinent. FEP tubing was used to connect all the liquid components. Above the arena, there was a camera for video recording/image analysis. Supplementary Fig. 6 shows an overall picture of the platform with the main parts highlighted.

**Droplet generation calibration.** Although the devices were always printed the same way, the final result was always slightly different. This difference was important in the case of the droplet generator outlet, as can be seen in Supplementary Fig. 3. In order to have a homogeneous droplet generation through all the devices with potential different outlet sizes, the speed at which the pulses were generated from the pumps was calibrated in order to obtain a perfect droplet generation when only 1-octanol was present in the mixture.

**Image processing and droplet detection.** A Microsoft LifeCam Cinema Web camera was situated above the arena in order to record the experiment. While the experiment happened, the camera stream was fed into a running Python (2.7.11) OpenCV (2.4.12) script, which performed the image analysis and returned the number of droplets active at the end of the experiment. The video was configured to 800 × 600 pixels, and 30 frames per second (FPS). The first step consisted of defining a circular area with 275 pixels of radius. This area overlapped with the experimental circular arena, and only the pixels inside this area were considered for image processing. The image processing was performed using a mixture of gaussians (MoG) model for background subtraction. OpenCV's MoG was used for this purpose using the default configuration values. The MoG model was reinitialised before each experiment. It is important to remark that everything in the scene remained constant except the oil droplets, therefore, all the pixels marked as foreground were droplets. The foreground subtracted was then used with a find contours operation from OpenCV in order to describe the droplets. At the end of the experiment, the number of droplets active was returned as a fitness value. By using a MoG with a small window, all the droplets that remained static a few seconds were considered as part of the background and discarded. This way, only the droplets that always moved were considered part of the foreground. See Supplementary Fig. 4 for pictures of how the different steps were performed.

**Lattice search.** Combinations of the four oils individually, in pairs, threes or fours with a granularity of 10% were tested in order to execute the lattice search. In the case of the oils being tested individually, only one experiment was performed, where one of the oils was active, and all the other ones were inactive. In all the other cases, a sequential and evenly spaced search was performed, where every step represented 10%. In this way, for example, in the case of two oils, there would be nine possible combinations (ignoring the extremes were only one of the oils is active): 0.1/0.9–0.2/0.8–0.3/0.7–0.4/0.6–0.5/0.5–0.6/0.4–0.7/0.3–0.8/0.2–0.9/0.1. The same procedure was applied to combinations of three or four oils. In total, there were 11 different combinations, six for pairs, four for triples and one for fours. Therefore, our lattice search consisted of 282 different oil formulations. As before, each formulation was tested five times, generating 1410 videos. Supplementary Fig. 7 shows the most interesting results.

**Genetic algorithm and fitness function.** A genetic algorithm was programmed using LabView 2015 standard libraries. Each GA run, except when described otherwise, consisted of 10 generations, and each generation had a population of 20 individuals. Each individual was defined by its recipe, which was the ratio between the four oils used (1-octanol, DEP, octanoic acid and 1-pentanol). Each of the oils was assigned a real number between 0 and 1, and the sum of the four oils for a given recipe was always 1. The first generation was constructed by assigning randomly generated recipes to each individual. Each individual was tested five times using the protocol described, and the average of these five executions was returned as its fitness value. Once all the individuals from a generation were given a fitness value, a new generation was constructed by choosing 10 parents from the just finished generation using the roulette wheel algorithm, where the individuals were selected with probability directly proportional to their fitness value. A given parent could only appear once in the following generation. The other 10 individuals were constructed by crossing the parents in randomly chosen pairs (each individual was chosen twice as parent), applying a random position one-point crossover, and a Gaussian 10% mutation (noise sample from a Gaussian distribution of mean 0 and variance 0.1). The final recipe was then normalised to 1.

**Fitness function.** Given a frame as provided by the camera stream, the droplets in that frame were detected using the algorithm described in the "Image processing and droplet detection" within this "Methods" section. Given an experiment, the fitness function calculated its fitness value as the number of droplets detected in the last frame of the provided camera stream. Because the experiments ran for 1 min, and we used 30 frames per second, this means that our fitness function can be exactly defined as the number of droplets detected by our algorithm in frame 1800. Because our droplet detection algorithm is based on a background subtraction

where we considered the droplets as foreground, if a droplet did not move for a set period of time (roughly 3 s) it was marked as part of the background, and removed from the count.

**Experimental reproducibility and control tests**. The result showed on Fig. 5 was repeated four times with similar results, see Supplementary Figs. 8, 9, 10 and 11. The same experiment but using an arena with pillars was repeated twice, see Supplementary Figs. 12 and 13. The same experiment but using an arena with a procedurally generated environment was repeated twice, see Supplementary Figs. 14 and 15. The experiment described in Fig. 6 was repeated once, with similar results, see Supplementary Fig. 16.

In order to study the drop in the evolutionary trajectories, the same hybrid GA run was performed but the first empty arena device was swapped by another empty arena device. In this case, the evolutionary trajectory kept a similar value, see Supplementary Figs. 20 and 21. The opposite experiment was also performed, where an initial GA run using the pillars arena was then swapped by one with the empty arena. The evolutionary trajectories grew slightly, see Supplementary Fig. 22. The experiment where an empty environment was swapped by one procedurally generated was also performed, see Supplementary Figs. 23 and 24. In both cases, the evolutionary trajectories dropped, but not as much as before. This reinforces the results seen on Fig. 6.

**Arena obstacles specification**. The pillars used both in the "pillars environment" and in the "cave environment" had a diameter of 2 mm. Their height was variable depending on their position, because the base of the arena has a slope. The shortest ones had a height around 4.7 mm, while the longest ones had a height of around 6.2 mm. In the "pillars environment", the pillars were placed in a grid, where each pillar had a neighbour in each of the possible four directions (north, south, east and west) when possible. The distance between pillars was 3 mm. See Supplementary Fig. 25.

The second custom arena used the same pillars as before, but instead of manually placing them in a linear/grid way, an algorithm was used to generate a pattern. The algorithm used was a Lindenmayer system. Examples of the patterns generated using this approach can be seen in Supplementary Fig. 26. The actual device used can be seen on Supplementary Fig. 27.

**Fitness landscape model**. The fitness landscapes shown in Fig. 6 were generated using support vector regression (SVR) with a radial basis function kernel. The experiment consisted of the described GA run where the 10 first generations used an empty environment, the following 10 used an environment populated with pillars and the last 10 used an environment with cave-like structures. For each of the three environments, data from a full GA run was collected, resulting in a data set of 200 experiments each. Because each experiment was repeated five times, each fitness landscape represents 1000 data points. To find the most accurate representation, best parameters for the SVR were estimated using 10-fold cross-validation and with respect to the mean squared error on the training set. Fitness landscapes are shown as ternary plot, a way to represent on a 2D plane three-coupled variables, which sum to a constant. In our case, we represent three out of the four components (with the fourth parameter held constant at zero). Our parameters represent percentage of each oils, the sum is thus constrained to 1. The C (penalty parameter of the error term) parameter was searched within [0.01, 0.1, 1, 10, 100]. The best parameters were C = 100 and gramma = 10. Supplementary Fig. 28 shows all the fitness landscapes produced this way with the collected data. The third row of this figure represents the fitness landscapes, which can be seen in Fig. 6.

Each one of the environments was also tested individually, meaning isolated GA runs for each environment where the first generation was generated randomly. Supplementary Fig. 29 shows the fitness landscapes generated with this data, which were also generated using a SVR. For each environment (empty, pillars and caves), data from two full GA runs were collated resulting in a data set of 400 experiments each. Because every experiment was repeated five times, 2000 points were used. The fitness landscapes were then generated using the method just described. For the empty environment, the best parameters were {'C': 100, 'gamma': 10} for an average mean square error of 6.83 (std = 4.00). For the pillars environment, the best parameters were {'C': 10, 'gamma': 100} for an average mean square error of 21.30 (std = 10.68). For the caves environment, the best parameters were {'C': 10, 'gamma': 100} for an average mean square error of 11.50 (std = 4.50).

In Supplementary Movie 3 describing the evolution of the fitness landscapes through the different environments, the fitness landscapes were calculated from the same data set as Fig. 6 in the main manuscript, but in this case the method used was kernel ridge regression with a radial basis function kernel as before. The main difference is that in this case the same kernel was used to calculate all the fitness landscapes. The parameters here used for the kernel were {'alpha': 10, 'gamma': 100}.

**Computer code availability**. All relevant computer code is available from the authors on reasonable request.

**Data availability**. All relevant data are available from the authors on reasonable request.

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

## Acknowledgements

We gratefully acknowledge financial support from the EPSRC for funding (Grant Nos. EP/H024107/1, EP/I033459/1, EP/J00135X/1, EP/J015156/1, EP/K021966/1, EP/K023004/1, EP/K038885/1, EP/L015668/1, EP/L023652/1), BBSRC (Grant No. BB/M011267/1), the EC (projects 610730 EVOPROG, 611640 EVOBLISS), L.C. thanks the Royal Society/Wolfson Foundation for a Merit Award and the ERC for an Advanced Grant (ERC-ADG, 670467 SMART-POM).

## Author contributions

L.C. conceived the initial concept and together with J.M.P. and S.T., L.C. wrote the paper with input from all the authors. J.M.P. and S.T. designed and built the 3D-printed fluidic chemorobotic platform, J.M.P. performed the experiments, J.T. designed the chemistry, both J.G. and J.M.P. analysed the data with help from A.H. and J.G. lead the team that helped do this work.

## Additional information

**Competing interests:** The authors declare no competing financial interests.

