## [Peer Review File · Nature Communications]

Reviewers' comments:

Reviewer #1 (Remarks to the Author):

GLOBAL ASSESSMENT

This manuscript reports an investigation carried out by means of an artificial evolution method that was developed with the aim of exploring the phenotypic plasticity of droplet protocells of variable composition in response to changes in the environment.

It is an extension of a general approach already conceived and published in Nat Comm three years ago (Parrilla-Gutierrez et al. 2014), but the novelty of the current work consists in focusing specifically on the role that environmental modifications have in the evolutionary dynamics of the population of droplets. For that purpose, the authors implement and analyze a set of artificial evolution experiments in which a sudden reconfiguration of the arena where the droplets move, change shape and interact takes place.

The article is interesting and constitutes a worth-pursuing line of research that contributes to open the field of protocell research, connecting it with artificial life, robotics, and evolutionary dynamics. However, I consider that several important criticisms and improvements need to be addressed by the authors in order for it to deserve publication in a journal like Nat Comm.

These, as explained in more detail below, should include:

(i) a better presentation of the approach and, in particular, an argument for its novelty with regard to other similar proposals carried out in the past within the field of 'autonomous' or 'evolutionary' robotics;

(ii) a more complete and accurate description/characterization of the dynamic behaviors and interactions of the droplets, specifying e.g., whether there are any fusion and growth events (if not, why?), or just movement and division, as stated in Fig. 2;

(iii) making explicit the actual fitness function that is being used in the experiments (beyond saying that it reflects 'droplet activity') and of the reasons behind that choice;

(iv) change of the Figures in which fitness values are said to be reported (portraying the 'number of droplets' in the 'y' axis is rather confusing);

(v) a deeper discussion on the idea of phenotypic plasticity and of the reasons why, despite the strong decoupling implemented through this methodology (i.e., the decoupling between the self-organizing/self-assembling dynamics of the droplets and their evolutionary fate – related to changes in their composition that are artificially introduced to serve an external fitness function) one could still claim that the system itself is "responding" (i.e., adapting) to the perturbations through "active" variations in its phenotype;

(vi) at least two additional sets of control experiments, related to the different arenas actually selected for the runs reported (selection for which no rationale is provided -- nor in the main text, nor in the SI): a.- experiments in an environment with a smaller number of pillars (to check/report on the influence of pillar density), and with a different L-system configuration (to check/report on the influence of the spatial distribution of the caves).

Unless all these critical points are properly addressed by the authors, I would not recommend publication.

MORE DETAILED CRITICISM

(i) Authors should argue (i.e., provide concrete reasons -- in the intro or the discussion sections) why their approach is more interesting or has more potential, from an evolutionary perspective, than those traditionally implemented in the field of so-called 'autonomous/embodied/evolutionary' robotics. This may look like a rather obvious requirement, given that the main feature being 'selected for' in their experiments is, basically, the mobility of the droplets (quite evident association to 'navigating robots'). But, surprisingly, no consideration on these lines is done. Nevertheless, the paper would clearly benefit from a comparison (no matter how brief) with some lines of research in the field of autonomous robotics in which embodiment was seriously taken into account in the past. After doing so, authors may also want to re-consider the convenience of keeping such strong claims or over-statements like "the impact of environmental changes on an artificially evolving system has never been studied outside of computer simulations", as expressed in their abstract.

(ii) The manuscript covers too succinctly the description of the basic behavior of each droplet and its potential interactions with other droplets of the population. For instance, no explanation is given about possible processes of aggregation (fusion) and growth (or, alternatively, about why these processes do not happen at all). On p. 3 it is mentioned that the composition mixtures were used to "cover a wide range of different polarities, densities, viscosities, solubility and possible interactions that may occur at the interface" but no further hints are given on how these variables (and concrete component combinations -- or formulations) provide phenotypic diversity at a 'one-droplet level' or in terms of 'droplet-to-droplet interactions' in the experiments. A number of references are introduced to convey the fact that droplets can show very complex dynamic behaviors... but then, does all this complexity unfold during the experiments carried out here? In other words, is Fig. 2 a crude simplification of the actual dynamics or a realistic one (for reasons that are not explained)?

(iii) Nowhere in the main text (nor in the SI) could I find an explicit account of the fitness function used in this work. If one resorts to ref. 17, it appears that there are, in fact, different ways of conceiving the fitness function in this context: the 'division fitness function', the 'movement fitness function' and the 'vibration fitness function'. What are the weights used here to measure 'droplet activity' and then perform the GA accordingly? Then, a rationale for the use of that fitness function should also be included. At some point (p. 5) it is stated: "A droplet was considered active if it moved and was bigger than a set threshold." Is motility so important for life/evolution (as it could be for other phenomena like 'cognition')? And is size somehow being taken into account, as well? Please, clarify this central issue.

(iv) Related to the previous point, the graphs in which the fitness values are reported are rather confusing, because they are portrayed in terms of 'number of droplets', as if this was the only important variable to follow in the evolution of the population. In fact, if the values in the 'y axis' were actually real number of droplets, at each generation they should 'reset automatically' to 20, shouldn't they?

(v) The concept of phenotypic plasticity relates, as the authors state in p.10, to the ability of an organism (with a given genotype) to modify its phenotype in response to changes in its environment. The artificial evolution methodology implemented in this article, however, pushes changes in the composition of the droplets through a much more indirect and decoupled procedure, which involves the GA, an externally defined fitness function and several generation steps. Droplets are not given the chance to change their formulation (and, thereby, their dynamics) "by themselves", so to speak -- it is

always through the GA. Therefore, if the analogy is kept, the discussion about it should be made much more carefully, acknowledging these potential difficulties of the approach, because the conditions and the pathway implemented here to reach "new/different phenotypes" are certainly not those of the natural case. As a matter of fact, phenotypic plasticity in real biology is more commonly associated with immobile organisms, like plants. Animals do not need so much of it, precisely, because they can adapt to environmental changes via movement -- something perhaps also worth commenting.

(vi) More concretely on the set of experiments carried out, it would be interesting to know to what extent specific features of the environments tried (e.g., the density of pillars in the second arena or the geometrical disposition of the caves in the third one) may have non-trivial effects on the results obtained. Although the authors might have checked this issue during the design of the work, they do not comment anything about this in the manuscript/SI. Therefore, two new controls are suggested: one in which the density of pillars is reduced significantly, and one in which the caves are arranged in a very different way (keeping their total number constant).

MINOR POINTS

Other less important corrections/suggestions for improvement have been indicated in the pdf attached. As for the SI, authors refer there to Figures of the main text that do not exist in the final submitted version (like Fig. 5 and 6).

Reviewer #2 (Remarks to the Author):

Who will be interested in reading the paper, and why?

I would recommend this paper to readers from a variety of scientific areas. The multidisciplinary research is presented. Because of the chemical evolutionary experiments and comparison of droplets to protocells, it is an interesting paper for people from artificial life community that focus on both experimental and theoretical studies of evolution / origin of life. It is an interesting paper also from engineering point of view because it (and the supplementary materials and the previous paper of this group – ref. 17) contains detailed description how to build and use a liquid handling robot for evolutionary experiments. Such a device is very modular and could be used also for other studies, e.g. biological.

What are the main claims of the paper and how significant are they?

Authors promise in the title of the paper the reading about "Artificial Evolution of Droplet Protocells in a Chemical Robot with Configurable Environments Leads to Phenotypic Plasticity" and it summarizes nicely the main points of the paper. Authors say that they can perform a chemical evolutionary experiment by using a robotic platform. It is shown that by changes of an environment in a simple system consisting of oil droplets it is possible to influence the droplet behavior and the droplets can adapt to environmental changes similarly as living entities.

How does the paper stand out from others in its field? Is the paper likely to be one of the five most significant papers published in the discipline this year?

To the best of my knowledge, there are some groups focusing on life-like behavior of droplets, some of them try to use robotic platforms for evolutionary experiments, however Cronin's group seems to be at least one step before them. I could say it is a very significant paper in this area this year.

Are the claims novel? If not, which published papers compromise novelty?

Authors already published a paper where they presented a chemorobotic platform for study of evolution of droplets (ref. 17). Anyway, I think present manuscript shows the progress of their

research and summarizes new interesting results. In comparison with their previous papers, here they publish the data about artificial evolution affected by environmental changes first time.

Are the claims convincing? If not, what further evidence is needed?

In my opinion, the paper is written clearly and the claims are convincing. Furthermore, the supplementary file is full of further information.

Are there other experiments or work that would strengthen the paper further? How much would further work improve it, and how difficult would this be? Would it take a long time?

The amount of experiments presented in this paper is sufficient. Supplements confirm the reproducibility of results.

Are the claims appropriately discussed in the context of previous literature?

The claims are appropriately discussed in the context of previous literature. Maybe, the list of references should be longer and some recent papers should be discussed (no paper from years 2017 even 2016..).

If the manuscript is unacceptable, is the study sufficiently promising to encourage the authors to resubmit? If the manuscript is unacceptable but promising, what specific work is needed to make it acceptable?

The manuscript is acceptable.

Is the manuscript clearly written? If not, how could it be made more clear or accessible to nonspecialists?

The paper is written clearly.

However, maybe some readers could be confused from the terminology. Here the authors use the term "chemical robot" for the robotic platform. There are papers using this term for objects, whose behavior is based on chemical principles and these objects are like "biomimetic robots". E.g. self-walking gels (Maeda, Shingo, et al. "Chemical robot—Design of self-walking gel—." *Intelligent Robots and Systems*, 2007. IROS 2007. IEEE/RSJ International Conference on. IEEE, 2007. Maeda, Shingo, et al. "Chemical robot-design of peristaltic polymer gel actuator." *Intelligent Robots and Systems*, 2009. IROS 2009. IEEE/RSJ International Conference on. IEEE, 2009.) or particles behaving like artificial cells (Grančič, Peter, and František Štěpánek. "Chemical Swarm Robots." *Handbook of Collective Robotics: Fundamentals and Challenges*. Pan Stanford, 2013. 745-771. Lagzi, István. "Chemical robotics—chemotactic drug carriers." *Central European Journal of Medicine* 8.4 (2013): 377-382.). I propose to use similar term as in previous papers of L.Cronin – liquid handling robot or chemorobotic platform (ref. 17, Hanczyc, Martin M., et al. "Creating and maintaining chemical artificial life by robotic symbiosis." *Artificial life* (2015).)

Would readers outside the discipline benefit from a schematic of the main result to accompany publication?

Considering the multidisciplinary research, readers from several disciplines would benefit from presented results. Moreover it is a cool topic – "artificial evolution of droplets in a dish".

Could the manuscript be shortened? (Because of pressure on space in our printed pages we aim to publish manuscripts as short as is consistent with a persuasive message.)

If needed, the authors could shorten the paper and move some parts to supplements.

Should the authors be asked to provide supplementary methods or data to accompany the paper

online? (Such data might include source code for modelling studies, detailed experimental protocols or mathematical derivations.)

Supplementary information is sufficient.

Have the authors done themselves justice without overselling their claims?

Yes.

Have they been fair in their treatment of previous literature?

Yes.

Have they provided sufficient methodological detail that the experiments could be reproduced?

Yes. The supplementary materials contain a lot of details.

Is the statistical analysis of the data sound, and does it conform to the journal's guidelines?

Yes.

Are the reagents generally available?

1-Octanol, diethyl phthalate (DEP), 1-pentanol, octanoic acid, and tetradecyltrimethylammonium bromide (TTAB) are reagents offered commonly by suppliers of chemicals. For example one of the largest chemical suppliers – Sigma-Aldrich – offers all of them. Also polypropylene for 3D printing is commonly available. 3D printer and other parts of the system are commercially available or the protocols for their fabrication available.

Are there any special ethical concerns arising from the use of human or other animal subjects?

There are no special ethical concerns arising from the use of human or other animal subjects.

Brief guide for submission to Nature Communications:

Title. If possible, this should be 15 words or fewer and should not contain technical terms, abbreviations, punctuation and active verbs.

The title of present paper does not fulfill this requirement.

Section headings should be used and subheadings may appear in 'Results'. Avoid 'Introduction' as a heading.

No headings and subheadings at all.

Methods. The Methods section appears in all online original research articles and should contain all elements necessary for interpretation and replication of the results.

The separate section 'Methods' is missing.

Figure legends should be <350 words each. Figure legends should begin with a brief title sentence for the whole figure and continue with a short statement of what is depicted in the figure, not the results (or data) of the experiment or the methods used. Legends should be detailed enough so that each figure and caption can, as far as possible, be understood in isolation from the main text.

I am not sure if all figures in this paper fulfill this requirement.

Legend of Figure 4 is longer than 350 words.

I don't like the structure TOP/ BOTTOM, left/middle/right, moreover with the combination of letters. Some figures seem confusing and it takes some time to orient in the figure and legend. E.g. Figure 3:

letters in the legend do not correspond to letters in the figure. Also the RIGHT division into further left and right is not well-arranged. In the text authors use “see Figure 3b”, although the legend and labels are different (no small letters in the figure). The legend in legend is not needed, write it directly into figure.

Figure 4. Don't use legend in the legend, write 1-pentanol, octanoic acid, DEP and 1-octanol directly in the figure. And again the division into top and bottom in such a long figure is not good. I highly recommend to improve the labels and legends of figures.

Manuscript – few comments / questions

Figure 2 – I miss the reference to the statement “While the oil surfactants were non-ionic, TTAB is a cationic surfactant, meaning that TTAB molecules crossed the semi permeable interface until equilibrium. At the same time, the different oils dissolved at different rates into the aqueous phase.” or further discussion in the text.

Page 5 - (see SI for behaviours b-e) – please, specify

Lidenmayer system¹⁹ (L-system) – somewhere authors use a L-system, somewhere an L-system.

Page 10 - ... only formed by lipid aggregates... ?

Author Information – twice the same paragraph.

(Btw. I am not a native English speaker, so I am unworthy to evaluate the English).

Supporting Information – few comments / questions

Authors use the name of the robotic platform “Flowbot” in the supporting information. This name is not appearing in the manuscript at all. Why? I randomly checked some grants that are acknowledged. For example project EVOBLISS, where L.Cronin is involved, has similar robot Evobot. I googled and I have seen in several projects some liquid handling robots. Are there any differences between these robots?

Page S1 SI-1 – use molar concentration (1 molar = 1 M = 1 mole/liter) instead of 10 mmol in 5 l etc.

Page S6 - Supplementary Figure 3 – better resolution?

Page S11 – “The experiment described in Figure 5 was repeated once,...” (and several times further) - there is no Figure 5 in the main manuscript.

Page S26 – “Supplementary Fig. 26 extends Figure 6 from the main manuscript...” – there is no Figure 6 in the main manuscript.

Page S28 – “Droplets alive at the end of experiment” – how is defined “alive” droplet?

Reviewer #3 (Remarks to the Author):

Authors report a concept of an evolution system with real droplets. This is not an incremental study but contains a conceptual novelty. The attempt appears to be interesting because the computer based concept is combined with real chemicals. However, it is difficult for me to evaluate the scientific significance of this paper. Following is my opinion.

Present manuscript is written for computer-based scientists and not comprehensive for chemists and physicists treating real chemicals. For example, each term of "generation", "environmental change", "evolution", "mutation", "crossover", "fitness" and etc., which is keyword for understanding this paper, is just a metaphor in droplet dynamics dealt in this study. The corresponding phenomenon that is observed in experiment should not be described by such metaphorical expressions. Instead, all real phenomena can be expressed by explicit (simpler) words that are used in chemistry and physics with real chemicals. Moreover, the mechanisms that possibly explain real droplet dynamics is not discussed. How do the four chemical components affect motility of a droplet? How does the genomic algorithm influence the chemical components of a droplet? Each experimental procedure is a concrete operation, and the resulting behavior should be understood in term of physical and chemical effects of the operation on droplet dynamics. Complete explanations are possible based on this scheme without metaphorical expressions. They should be provided after the explicit scientific descriptions. I feel that this study may contain conceptual novelty with scientific impact. However, the present form of manuscript does not provide multidisciplinary and broad readability. In my opinion, the present form is not appropriate for Nature communications.

Reviewer #4 (Remarks to the Author):

The paper considers the application of artificial evolutionary techniques to droplet protocells and in particular considers the effects of changing the environment in which the evolution take place and whether interesting properties of the resulting phenotypes can be observed.

The novelty does not appear to lay in the evolutionary processes described in the paper, this aspects seems to be fairly standard genetic algorithm techniques and processes commonly used in many artificial population-based evolutionary set ups. The application to droplet protocells is however novel as far as this review is aware.

The paper states that "the impact of environmental changes on an artificially evolving system has never been studied outside of computer simulations" – I believe that "never" in this context is too strong. Early work by Thompson, for example, considered evolution on electronic hardware (not simulation) and did consider environmental changes (such as temperature and radiation), work in the early to mid 90's by JPL considered significant changes in environmental conditions with a for of field programmable transistor array (similar work was also conducted around the same time at the University of Heidelberg. More recently work related to the EU project SYMBRION consider evolution on real robots in the presence of changing environments. While I am not saying this is not an interesting experiment to carry out, I do not believe "never" is correct in this context.

The paper states that "The objective was to create droplets that would provide a chemical compartment related to the aforementioned [which included: different polarities, densities, viscosities, solubility and interactions] attributes, aiming for their stability and the ability to move and divide". The actual fitness used to drive evolution in these experiments appears to have been "to maximise the number of active droplets after one minute of observation time". It is not at all clear to me how this

objective function (fitness function) can be used to measure let alone guide evolution to achieve these stated objectives?

A number of “surprising and unexpected behaviours” were observed and listed in the paper; I am not an expert in droplet behaviours so could not really comment on whether these behaviours were surprising or not, but I do find it difficult to see the relevance of these to the artificial evolution that is the focus of the paper?

It would have been interesting to have more analysis on how the artificial evolution change between having a stable environment and a changing environment? Were the individuals evolved during the changing environment experiments in some way more adaptive to environments they had not been evolved in than the stable individuals? Did they divide more? Did any of the attributes mentioned before change?

Environment is being used as an evolutionary pressure on the process. One would expect a GA (or other evolutionary mechanism) to produce different individuals when faced with a changed environment – that is what evolution does. Not really sure this is an unexpected result. What happens if you put a good individual from the L-system back into the empty arena, does it produce a good individual quicker now than when it evolved from scratch – has this exposure to various environments made individuals more adaptive?

The paper states “This research also highlights that engineering the environment itself as a variable is a promising approach for the optimization of complex systems.” It would be interesting if this statement was expanded somewhat and examples given as to what the aim of doing this might be and where it could be applicable.

I do not think the title of the paper properly describes what the essence of the paper is about – “Chemical Robot” are there really robots in these experiments, chemical or otherwise? Surely robots are entities that perform a series of “complex” actions, I don’t really see any actions going on here.

Why does the evolution use number of active droplets as its fitness function? And why is a one minutes observation window appropriate?

Having a GA run 10 generations and a population of 20 individuals is quite small, I wonder if larger populations and more generations were ever used to see if the evolution had in fact stopped?

I actually like the experiments outlined in this paper and think it could well be of interest to many researchers in a number of different fields. But for me at the moment it raises too many unanswered questions.

We thank the reviewers for their valuable comments. We incorporated them in the paper and address them point by point in the following text. The referee comments are in bold, and our replies in normal type.

Reviewer #1 (Remarks to the Author):

GLOBAL ASSESSMENT

This manuscript reports an investigation carried out by means of an artificial evolution method that was developed with the aim of exploring the phenotypic plasticity of droplet protocells of variable composition in response to changes in the environment.

It is an extension of a general approach already conceived and published in Nat Comm three years ago (Parrilla-Gutierrez et al. 2014), but the novelty of the current work consists in focusing specifically on the role that environmental modifications have in the evolutionary dynamics of the population of droplets. For that purpose, the authors implement and analyze a set of artificial evolution experiments in which a sudden reconfiguration of the arena where the droplets move, change shape and interact takes place.

The article is interesting and constitutes a worth-pursuing line of research that contributes to open the field of protocell research, connecting it with artificial life, robotics, and evolutionary dynamics. However, I consider that several important criticisms and improvements need to be addressed by the authors in order for it to deserve publication in a journal like Nat Comm.

We would like to thank the reviewer for positive and detailed comments and for understanding the novelty of our research.

These, as explained in more detail below, should include:

Because this reviewer divided her/his comments into first a concise version, and then a more detailed description, we have copied here both versions together, in order to only answer each point once, trying to address both the quick snippet and the more detailed question at the same time.

(i) a better presentation of the approach and, in particular, an argument for its novelty with regard to other similar proposals carried out in the past within the field of 'autonomous' or 'evolutionary' robotics;

Authors should argue (i.e., provide concrete reasons -- in the intro or the discussion sections) why their approach is more interesting or has more potential, from an evolutionary perspective, than those traditionally implemented in the field of so-called 'autonomous/embodied/evolutionary' robotics. This may look like a rather obvious requirement, given that the main feature being 'selected for' in their experiments is, basically, the mobility of the droplets (quite evident association to 'navigating robots'). But, surprisingly, no consideration on these lines is done. Nevertheless, the paper would clearly benefit from a comparison (no matter how brief) with some lines of research in the field of autonomous robotics in which embodiment was seriously taken into account in the past. After doing so, authors may also want to re-consider the convenience of keeping such strong claims or over-statements like "the impact of environmental changes on an artificially evolving system has never been studied outside of computer simulations", as expressed in their abstract.

We have added the word "chemical" in the abstract, and now the sentence reads like "the impact of environmental changes on an artificial chemical evolving system has never been studied outside of computer simulations". We have changed the word "never" to "rarely" in the first paragraph.

We do agree with this reviewer that this research might be of interest to the field of evolutionary or autonomous robotics, but because we are a chemistry group which focuses on topics such as the origin of life and 3D printing, that is the angle we decided to use in this research. We also agree on the idea of considering oil droplets as “robots”, and therefore studying them in a similar context, but we considered that the leap from “droplet” to “robot” might be too big for the community to accept. We have expanded the introduction addressing the points expressed by this reviewer focusing on the concept of bio-inspired robotics.

The main difference between our system and the field of evolutionary robotics is that our robots are purely wet, just chemical entities, and thus they don't contain any mechanical, mobile or electronic parts. Thus, that while in most of the research of evolutionary robotics the evolution happens “in silico” (each robot has some sort of microcontroller which is re-programmed as the evolution goes on), in our system part of the evolution happens “in silico”, but part of it happens “in vitro”. Our droplets are not reprogrammed, but created from scratch before each iteration. It is also important to note that while most of the work in evolutionary robotics uses “evolution” as an optimization mean, we used it both as an optimization mean, but also with the idea of investigating how in a origin of life context, simple lipid aggregates could use biological (or chemical) evolution in order to acquire new behaviours, and in particular in this research we focused on how the environment helped that evolutionary process. We do consider that our work has more in common with other chemistry research that used algorithms for optimization or discovery (such as the work of Bawazer *et al.* which has now been added as a reference in the manuscript) or the work of Ismagilov *et al.* in JACS 2010, 132(9), than with traditional evolutionary robotics.

(ii) a more complete and accurate description/characterization of the dynamic behaviors and interactions of the droplets, specifying e.g., whether there are any fusion and growth events (if not, why?), or just movement and division, as stated in Fig. 2;

The manuscript covers too succinctly the description of the basic behavior of each droplet and its potential interactions with other droplets of the population. For instance, no explanation is given about possible processes of aggregation (fusion) and growth (or, alternatively, about why these processes do not happen at all).

There are no fusion or growth events in our system. As far as we are aware, there are no growth or fusion behaviours reported in the literature when using similar systems to ours. Surfactant molecules are actively fighting against fusion events. The only reported paper where a fusion was directly explained is the work of Caschera *et al.* (referenced in the manuscript) where they used salt to weaken the surfactant interface, to the point where the droplets fused. In our system we did not use salts. There is also the work of Banno *et al.* [1], where they achieved droplet fusion as a side effect, but their system was very different to ours.

There is no growth because the droplets are in an aqueous medium, in a ratio of 10 μ L droplets in 2 ml of aqueous phase. The gradients go from inside to outside, meaning that the droplets dissolve their matter into the aqueous phase, shrinking through time. This is very similar to the self propelled droplets of Hanczyc *et al.* which we have referenced within the main manuscript.

Based on the literature, thus, fusion and growth were not expected to happen, and in fact they did not happen during our experiments.

[1] Mode changes associated with oil droplet movement in solutions of gemini cationic surfactants. Banno *et al.* *Langmuir*, **29**, 7689-7696 (2013)

On p. 3 it is mentioned that the composition mixtures were used to "cover a wide range of different polarities, densities, viscosities, solubility and possible interactions that may occur at the interface" but no further hits are given on how these variables (and concrete component combinations -- or formulations) provide phenotypic diversity at a 'one-droplet level' or in terms of 'droplet-to-droplet interactions' in the experiments.

We based ourselves in our previously published literature (Gutierrez et al. Nat Comm 2014), because our work here is not about the behaviours per se, but about the interplay between behaviours, evolution and environment. The phenotypic plasticity of the system is shown when the different environments are tested, because for example in the previous research also published in Nat Comm, using a very similar chemical system, we never observed back then this kind of phenomena.

A number of references are introduced to convey the fact that droplets can show very complex dynamic behaviors... but then, does all this complexity unfold during the experiments carried out here?

The key aspect of this work is to have a simple chemical system in terms of the components, but at the same time can be capable of showing complex phenomena. The complexity in terms of behaviours is interesting, but not the subject of this work. We will be exploring this in further work.

Here, the key point was to have a complex system that could be shaped by the environment. As such, we studied our system from a "system chemistry" perspective, where we studied the system as a whole and studied how the whole system changed by modifying its variables. It can be said that we followed a "top-bottom" approach in the way we tried to discover new droplet behaviours, considering the chemistry a "black box" and letting an algorithm without any kind of chemical knowledge to study it, instead of focusing on the molecules interactions, and studying how those interactions can affect the system.

In other words, is Fig. 2 a crude simplification of the actual dynamics or a realistic one (for reasons that are not explained)?

Figure 2 describes the key interactions that govern the behaviours suitable for a general audience, so it is simplistic but that does not detract from the key aim of this study – the effect of the environment on evolving chemical systems.

(iii) making explicit the actual fitness function that is being used in the experiments (beyond saying that it reflects 'droplet activity') and of the reasons behind that choice;

Nowhere in the main text (nor in the SI) could I find an explicit account of the fitness function used in this work. If one resorts to ref. 17, it appears that there are, in fact, different ways of conceiving the fitness function in this context: the 'division fitness function', the 'movement fitness function' and the 'vibration fitness function'. What are the weights used here to measure 'droplet activity' and then perform the GA accordingly?

We think that the fitness function is clearly stated in the main manuscript as follows:

This was done using a fitness function that aimed to maximise the number of active droplets after one minute of observation time. A droplet was considered active if it moved and was bigger than a set threshold. Therefore, experiments where the droplets underwent controlled division and movement were assigned correspondingly high fitness values.

SI-4 explains how the droplets were detected. Thus using the method described there, the fitness value of an experiment is the number of droplets detected in the last frame of an experiment. If we

consider than we run the experiment for a minute, and that every second had 30 frames, an experiment in total had 1800 frames, thus the fitness value was the number of droplets detected in the 1800th frame. Because we use a background subtraction algorithm to detect the droplets, this value depended on the previous frames. If a droplet did not move for a certain period (this depends in the background subtraction algorithm implementation, in our case it was roughly 3 seconds), the algorithm then considers the droplet as part of the background, and thus it is removed from the count.

During this research we only used one fitness function, which is the one stated above. In previous research we used, as this reviewer said, three different functions. None of those fitness functions were used here. It can be said that the fitness function used here is a mixture of what we called in previous research “movement” and “division”. The main difference is that now we do not care about how much a droplet moves, as long as it moves.

To clarify the above points, we have expanded the information of the fitness function in the SI (SI-8).

The image processing and the fitness function were coded in Python using the OpenCV library for computer vision. In our code the fitness function is simply described as:

```
return len(droplets_per_frame[-1])
```

Where `droplets_per_frame` is a data structure which contains as many elements as frames, and in each element it contains the X and Y center position of each droplet detected. Therefore, by indexing the “-1” position we are returning the last element (or frame), and by using the “len” function we are returning its size. This is literally the number of droplets at the end of the experiment.

The size parameter being taken into account is just a discriminant used in the computer vision algorithm in order to separate from droplets to noise. In our case, for a particle to be a droplet it had to contain at least 2 pixels. Although this threshold might seem very low, the number of false positives was near 0.

This information has also been added to the SI.

Then, a rationale for the use of that fitness function should also be included. At some point (p. 5) it is stated: "A droplet was considered active if it moved and was bigger than a set threshold."

The reason for choosing this fitness function is because as said, in previous research we used among other functions “movement” and “division”. We wanted to use a fitness function which was similar enough in order to validate our platform against the results obtained before. This research focuses on the impact of the environment, therefore we decided to keep said fitness function and focus instead on the different devices and their designs.

Is motility so important for life/evolution (as it could be for other phenomena like 'cognition')?

Yes, we think so, although the choice of “movement” and “division” behaviour in this study was more a metaphorical reason. Motility may not be the essential property for an (chemical/biology) entity to be “alive”; however, we think it is essential for protocells to evolve further because motility renders them more dynamic complexity and also allow them to, for example, migrate to better environment or hunt for energy source. A similar point was discussed in the context of Artificial Life system by Ikegami et al. (*BioSystems*, 91, 388–400, 2009).

And is size somehow being taken into account, as well? Please, clarify this central issue.

Yes, please refer to our response above.

(iv) change of the Figures in which fitness values are said to be reported (portraying the 'number of droplets' in the 'y' axis is rather confusing);

We are sorry for the confusing labelling of the graphs. We labelled the Y axis as “number of droplets”, because the fitness function (which is the Y axis) describes “the number of actively moving droplets at the end of the experiment”. In any case, following the advice of this reviewer, we have changed the Y label in what is now Figure 5 and 6 from “number of droplets” to “fitness value” to avoid confusion.

Related to the previous point, the graphs in which the fitness values are reported are rather confusing, because they are portrayed in terms of 'number of droplets', as if this was the only important variable to follow in the evolution of the population. In fact, if the values in the 'y axis' were actually real number of droplets, at each generation they should 'reset automatically' to 20, shouldn't they?

In the graphs shown, we described the average fitness value per generation. The fitness value associated to a recipe is the averaged fitness value of 5 independent repeats of a droplet experiment using that recipe. Each generation contained 20 recipes, so the average value of these 20 recipes is the number portrayed in our graphs. This point should now be clarified in the manuscript.

(v) a deeper discussion on the idea of phenotypic plasticity and of the reasons why, despite the strong decoupling implemented through this methodology (i.e., the decoupling between the self-organizing/self-assembling dynamics of the droplets and their evolutionary fate – related to changes in their composition that are artificially introduced to serve an external fitness function) one could still claim that the system itself is "responding" (i.e., adapting) to the perturbations through "active" variations in its phenotype;

First, we would say that the reviewer's view on decoupling between droplet dynamics and evolutionary fate is probably not possible – in fact we are looking for a system that actively couples them, as we think this is important for the origin of life. In our droplet system, the self-organizing dynamics of droplets (i.e. maintaining a droplet, division into multiple droplets, interaction with external aqueous phase at the interface, dissolution of droplet components into the aqueous phase, etc) determines the fate of evolution. Thus, we do not see any decoupling here.

The concept of phenotypic plasticity relates, as the authors state in p.10, to the ability of an organism (with a given genotype) to modify its phenotype in response to changes in its environment. The artificial evolution methodology implemented in this article, however, pushes changes in the composition of the droplets through a much more indirect and decoupled procedure, which involves the GA, an externally defined fitness function and several generation steps. Droplets are not given the chance to change their formulation (and, thereby, their dynamics) "by themselves", so to speak -- it is always through the GA. Therefore, if the analogy is kept, the discussion about it should be made much more carefully, acknowledging these potential difficulties of the approach, because the conditions and the pathway implemented here to reach "new/different phenotypes" are certainly not those of the natural case. As a matter of fact, phenotypic plasticity in real biology is more commonly associated with immobile organisms, like plants. Animals do not need so much of it, precisely, because they can adapt to environmental changes via movement -- something perhaps also worth commenting.

Our use of the term was more inspired from bacterial phenotypic plasticity where bacteria are capable of coping with various environments by varying phenotypic behaviour (e.g. gene expressions) despite of no changes to genetic composition. Similarly, we meant to imply droplets capable of coping with

different environments (e.g. empty and pillar environments). This can be seen, for example, in Fig. 6 ternary plot for fitness where recipes in the lower left region show relative high fitness in the first (empty) and third (L-system) environments. However we can see this term might cause problems, hence we have changed it to adaptive.

We also think it is important to note that some well-defined biological concepts like “phenotypic plasticity” or “speciation” had to have a simpler mechanism in the context of the origin of life, because the life forms at the very beginning were simpler than most of the life forms known nowadays. Thus, while the concept of phenotypic plasticity is associated nowadays to how plants change their attributes based on the season, simpler protocells back then had to have similar but simpler phenomena in order to survive. Speciation, for example, had to happen when life was no more than a basic protocell, and that speciation was simpler to the known nowadays, because we are speaking of basic protocells formed by an aggregation of lipids. We think it is important to extrapolate these concepts to the scenario happening around 4 billions of years ago.

However, to avoid confusion, we changed the term as “adaptability” to be more general and also to avoid confusion. This change is now reflected in the manuscript.

(vi) at least two additional sets of control experiments, related to the different arenas actually selected for the runs reported (selection for which no rationale is provided -- nor in the main text, nor in the SI): a.- experiments in an environment with a smaller number of pillars (to check/report on the influence of pillar density), and with a different L-system configuration (to check/report on the influence of the spatial distribution of the caves).

More concretely on the set of experiments carried out, it would be interesting to know to what extent specific features of the environments tried (e.g., the density of pillars in the second arena or the geometrical disposition of the caves in the third one) may have non-trivial effects on the results obtained. Although the authors might have checked this issue during the design of the work, they do not comment anything about this in the manuscript/SI. Therefore, two new controls are suggested: one in which the density of pillars is reduced significantly, and one in which the caves are arranged in a very different way (keeping their total number constant).

Although we do agree that follow-up experiments are very interesting, and we plan to do so in the future, we do not think that more experiments are necessary in order to address the main hypothesis of our research. The key thesis of this work is “does the environment have an impact on the evolution of our droplet system?”. We think that not only have we enough data to prove this point, including all the control experiments that are part of the SI, we have also shown that simple mixtures of chemicals can have complex behaviours that can couple to the environment. The data collected for this research required 19,350 single experiments (or approximately 100,000 droplet measurements), and it took roughly two years to obtain it. In future work we are developing a new platform to expand as the referee suggests above, but we are excited by key findings presented here and hope that we can convince the referee of the urgent need to report these findings in this communication.

Unless all these critical points are properly addressed by the authors, I would not recommend publication.

We feel we have addressed all of the points of the referee and thank them for their constructive and interesting critique.

MINOR POINTS

Other less important corrections/suggestions for improvement have been indicated in the pdf attached. As for the SI, authors refer there to Figures of the main text that do not exist in the final submitted version (like Fig. 5 and 6).

[Editor's note: Please also see the attached annotated files, which were provided by Reviewer #1]

We really appreciate the exceptional feedback given by this reviewer in the documents attached. We have done almost all the changes suggested. In particular, we removed a sentence where we were talking about the open ended evolution possibilities of our platform.

Reviewer #2 (Remarks to the Author):

Who will be interested in reading the paper, and why?

I would recommend this paper to readers from a variety of scientific areas. The multidisciplinary research is presented. Because of the chemical evolutionary experiments and comparison of droplets to protocells, it is an interesting paper for people from artificial life community that focus on both experimental and theoretical studies of evolution / origin of life. It is an interesting paper also from engineering point of view because it (and the supplementary materials and the previous paper of this group – ref. 17) contains detailed description how to build and use a liquid handling robot for evolutionary experiments. Such a device is very modular and could be used also for other studies, e.g. biological.

First of all, we would like to thank the reviewer for understanding the novelty of our study. We addressed all the points raised by the reviewer. We now feel the manuscript has improved significantly.

What are the main claims of the paper and how significant are they? Authors promise in the title of the paper the reading about “Artificial Evolution of Droplet Protocells in a Chemical Robot with Configurable Environments Leads to Phenotypic Plasticity” and it summarizes nicely the main points of the paper. Authors say that they can perform a chemical evolutionary experiment by using a robotic platform. It is shown that by changes of an environment in a simple system consisting of oil droplets it is possible to influence the droplet behavior and the droplets can adapt to environmental changes similarly as living entities.

How does the paper stand out from others in its field? Is the paper likely to be one of the five most significant papers published in the discipline this year?

To the best of my knowledge, there are some groups focusing on life-like behavior of droplets, some of them try to use robotic platforms for evolutionary experiments, however Cronin’s group seems to be at least one step before them. I could say it is a very significant paper in this area this year.

We are happy that the referee is excited about our work and its impact.

Are the claims novel? If not, which published papers compromise novelty? Authors already published a paper where they presented a chemorobotic platform for study of evolution of droplets (ref. 17). Anyway, I think present manuscript shows the progress of their research and summarizes new interesting results. In comparison with their previous papers, here they publish the data about artificial evolution affected by environmental changes first time.

Are the claims convincing? If not, what further evidence is needed? In my opinion, the paper is written clearly and the claims are convincing. Furthermore, the supplementary file is full of further information.

Are there other experiments or work that would strengthen the paper further? How much would further work improve it, and how difficult would this be? Would it take a long time? The amount of experiments presented in this paper is sufficient. Supplements confirm the reproducibility of results.

Are the claims appropriately discussed in the context of previous literature? The claims are appropriately discussed in the context of previous literature. Maybe, the list of references should be longer and some recent papers should be discussed (no paper from years 2017 even 2016.).

We have added a new reference from 2016, and another one from 2017 which was just published while this paper was reviewed. We are open to any other recommendation from this reviewer.

If the manuscript is unacceptable, is the study sufficiently promising to encourage the authors to resubmit? If the manuscript is unacceptable but promising, what specific work is needed to make it acceptable? The manuscript is acceptable.

Is the manuscript clearly written? If not, how could it be made more clear or accessible to nonspecialists? The paper is written clearly.

However, maybe some readers could be confused from the terminology. Here the authors use the term “chemical robot” for the robotic platform. There are papers using this term for objects, whose behavior is based on chemical principles and these objects are like “biomimetic robots”. E.g. self-walking gels (Maeda, Shingo, et al. "Chemical robot—Design of self-walking gel—." Intelligent Robots and Systems, 2007. IROS 2007. IEEE/RSJ International Conference on. IEEE, 2007. Maeda, Shingo, et al. "Chemical robot-design of peristaltic polymer gel actuator." Intelligent Robots and Systems, 2009. IROS 2009. IEEE/RSJ International Conference on. IEEE, 2009.) or particles behaving like artificial cells (Grančič, Peter, and František Štěpánek. "Chemical Swarm Robots." Handbook of Collective Robotics: Fundamentals and Challenges. Pan Stanford, 2013. 745-771. Lagzi, István. "Chemical robotics—chemotactic drug carriers." Central European Journal of Medicine 8.4 (2013):377-382.).

I propose to use similar term as in previous papers of L.Cronin – liquid handling robot or chemorobotic platform (ref. 17, Hanczyc, Martin M., et al. "Creating and maintaining chemical artificial life by robotic symbiosis." Artificial life (2015).)

As this reviewer explains, we used the term “chemical robot” to refer to the robotic platform. We have changed the wording from “chemical robot” to “3D printed fluidic chemorobotic platform” both in the title and in the abstract, and clarified it everywhere else in the main manuscript.

Would readers outside the discipline benefit from a schematic of the main result to accompany publication? Considering the multidisciplinary research, readers from several disciplines would benefit from presented results. Moreover it is a cool topic – “artificial evolution of droplets in a dish”.

Could the manuscript be shortened? (Because of pressure on space in our printed pages we aim to publish manuscripts as short as is consistent with a persuasive message.) If needed, the authors could shorten the paper and move some parts to supplements.

We feel that the ideas expressed in the manuscript are well served by all the content that it contains but would be open to requests from the editor for some location changes. However the key aspect, with such a cross-disciplinary research work, is to ensure that the readers understand and can reproduce the work.

Should the authors be asked to provide supplementary methods or data to accompany the paper online? (Such data might include source code for modelling studies, detailed experimental protocols or mathematical derivations.) Supplementary information is sufficient.

Have the authors done themselves justice without overselling their claims?

Yes.

Have they been fair in their treatment of previous literature?

Yes.

Have they provided sufficient methodological detail that the experiments could be reproduced?

Yes. The supplementary materials contain a lot of details.

Is the statistical analysis of the data sound, and does it conform to the journal's guidelines?

Yes.

Are the reagents generally available?

1-Octanol, diethyl phthalate (DEP), 1-pentanol, octanoic acid, and tetradecyltrimethylammonium bromide (TTAB) are reagents offered commonly by suppliers of chemicals. For example one of the largest chemical suppliers – Sigma-Aldrich – offers all of them. Also polypropylene for 3D printing is commonly available. 3D printer and other parts of the system are commercially available or the protocols for their fabrication available.

Are there any special ethical concerns arising from the use of human or other animal subjects?
There are no special ethical concerns arising from the use of human or other animal subjects.

Brief guide for submission to Nature Communications:

Title. If possible, this should be 15 words or fewer and should not contain technical terms, abbreviations, punctuation and active verbs.

The title of present paper does not fulfill this requirement

The title has been reviewed and it is now 15 words long.

Section headings should be used and subheadings may appear in 'Results'. Avoid 'Introduction' as a heading.

No headings and subheadings at all.

We have added section headings and subheadings.

Methods. The Methods section appears in all online original research articles and should contain all elements necessary for interpretation and replication of the results. The separate section 'Methods' is missing.

Currently the methods section is part of the SI. We can move it to the main manuscript if it is required by the editor.

Figure legends should be <350 words each. Figure legends should begin with a brief title sentence for the whole figure and continue with a short statement of what is depicted in the figure, not the results (or data) of the experiment or the methods used. Legends should be detailed enough so that each figure and caption can, as far as possible, be understood in isolation from the main text. I am not sure if all figures in this paper fulfill this requirement.

Legend of Figure 4 is longer than 350 words.

I don't like the structure TOP/ BOTTOM, left/middle/right, moreover with the combination of letters. Some figures seem confusing and it takes some time to orient in the figure and legend. E.g. Figure 3: letters in the legend do not correspond to letters in the figure. Also the RIGHT division into further left and right is not well-arranged. In the text authors use "see Figure 3b", although the legend and labels are different (no small letters in the figure). The legend in legend is not needed, write it directly into figure.

Figure 4. Don't use legend in the legend, write 1-pentanol, octanoic acid, DEP and 1-octanol directly in the figure. And again the division into top and bottom in such a long figure is not good. I highly recommend to improve the labels and legends of figures.

We have split the figures into single subfigures, also splitting the captions into their respective portions. The manuscript now has 7 figures. None of the new captions go over 350 words. By doing so we also cleared the structure of top/bottom, left/middle/right.

We have divided Figure 3 into two different figures, which are now Figure 4 and Figure 5. We hope this makes it clearer.

We have divided Figure 4 into 2 different figures, which are now Figure 6 and Figure 7. As before, we hope this makes it clearer.

Manuscript – few comments / questions

Figure 2 – I miss the reference to the statement "While the oil surfactants were non-ionic, TTAB is a cationic surfactant, meaning that TTAB molecules crossed the semi permeable interface until equilibrium. At the same time, the different oils dissolved at different rates into the aqueous phase." or further discussion in the text.

We have added some extra information in the caption. The mechanism we are referring here to is the one already reported by Caschera *et al.* (which is already referenced within the main text): If we have a system where in its initial state surfactants with a given charge are in a phase A, and surfactants with another charge B, these both surfactants will cross the interface until equilibrium.

Page 5 - (see SI for behaviours b-e) – please, specify

We have added a Figure which shows these behaviours and that we forgot to include before.

Lidenmayer system¹⁹ (L-system) – somewhere authors use a L-system, somewhere an L-system.

This should be fixed now.

Page 10 - ... only formed by lipid aggregates... ?

We have changed "aggregates" with "molecules".

Author Information – twice the same paragraph.

This has been fixed now.

(Btw. I am not a native English speaker, so I am unworthy to evaluate the English).

Supporting Information – few comments / questions

Authors use the name of the robotic platform “Flowbot” in the supporting information. This name is not appearing in the manuscript at all. Why? I randomly checked some grants that are acknowledged. For example project EVOBLISS, where L.Cronin is involved, has similar robot Evobot. I googled and I have seen in several projects some liquid handling robots. Are there any differences between these robots?

We are sorry for this mistake. Flowbot was the codename of this project, and now this term has been fully removed from the SI.

Evobot and Flowbot are very different platforms. Evobot is an automated liquid handling robot, with a mobile tool. It uses strut profiles, hardware components, motors, belts, pulleys. Flowbot is a fully 3D printed integrated platform. Flowbot is more similar to a microfluidic device than to a liquid handling robot. Its operation is more similar to flow chemistry than to the standard batch chemistry, which is what project like Evobot does.

Page S1 SI-1 – use molar concentration (1 molar = 1 M = 1 mole/liter) instead of 10 mmol in 5 l etc.

We have decided to remove the molar concentration. We consider that giving the mass is already enough information to repeat our experiments.

Page S6 - Supplementary Figure 3 – better resolution?

We are sorry to say that this is the only picture we have. In any case, even at low resolution it does show how the same 3D model printed in the same 3D-printer can produce slightly different results.

Page S11 – “The experiment described in Figure 5 was repeated once,...” (and several times further) - there is no Figure 5 in the main manuscript.

This has been fixed now. Thank you for the feedback.

Page S26 – “Supplementary Fig. 26 extends Figure 6 from the main manuscript...” – there is no Figure 6 in the main manuscript.

This has been fixed now. Thank you for the feedback.

Page S28 – “Droplets alive at the end of experiment” – how is defined “alive” droplet?

This has been changed to “active”. During the first drafts we called our fitness function as counting the number of droplets alive at the end of the experiment, but this has been changed to active through the whole manuscript and SI.

Reviewer #3 (Remarks to the Author):

Authors report a concept of an evolution system with real droplets. This is not an incremental study but contains a conceptual novelty. The attempt appears to be interesting because the computer

based concept is combined with real chemicals. However, it is difficult for me to evaluate the scientific significance of this paper. Following is my opinion.

Present manuscript is written for computer-based scientists and not comprehensive for chemists and physicists treating real chemicals. For example, each term of "generation", "environmental change", "evolution", "mutation", "crossover", "fitness" and etc., which is keyword for understanding this paper, is just a metaphor in droplet dynamics dealt in this study. The corresponding phenomenon that is observed in experiment should not be described by such metaphorical expressions. Instead, all real phenomena can be expressed by explicit (simpler) words that are used in chemistry and physics with real chemicals.

Although we have tried to ensure that our research was not written for computer-based scientist, but for a general audience, we appreciate the feedback. We used these terms because they are not metaphors, but commonly used terminologies in the context of Artificial Intelligence, in particular, Genetic Algorithms. These terms were not used in our manuscript to describe the droplet behaviour, but to describe how the algorithm and platform works. Moreover, we are not the only research group using similar terms in the context of chemistry of physics studies, as shows for example the recent work of Bawazer et al., reference which has been added to the main manuscript.

Moreover, the mechanisms that possibly explain real droplet dynamics is not discussed. How do the four chemical components affect motility of a droplet? How does the genomic algorithm influence the chemical components of a droplet? Each experimental procedure is a concrete operation, and the resulting behavior should be understood in term of physical and chemical effects of the operation on droplet dynamics. Complete explanations are possible based on this scheme without metaphorical expressions. They should be provided after the explicit scientific descriptions.

The mechanisms that explain the droplet behaviours seen in our research (division and movement) have been widely been reported in the literature. For example, Caschera *et al.* – referenced in the main manuscript- described how by using different concentrations of surfactant molecules inside and outside a droplet, division can be induced. And for example, Hanczyc *et al.* – also in the main manuscript- described how droplets can achieve motion via Marangoni instabilities. More precisely, for example, in a paper which was published while this paper was reviewed, Mass *et al.* described how using a system very similar to ours – they used the same TTAB surfactant molecule- droplet motion could be achieved. This reference has been added to the main manuscript.

The objective of our research was not to explain these already reported behaviours, but on one hand to show how an algorithm can discover them without any kind of chemical knowledge, and on the other hand to link these basic behaviours to abiogenesis, and see if very basic lipid aggregates could learn them, and moreover, study how changes in the environment could impact this evolutionary process. Our approach is more similar to systems chemistry than to standard synthetic chemistry. In future work we will dig into the mechanism of the chemical behaviours.

I feel that this study may contain conceptual novelty with scientific impact. However, the present form of manuscript does not provide multidisciplinary and broad readability. In my opinion, the present form is not appropriate for Nature communications.

Thank you for understanding the novelty of our study, and we have tried to make changes to address the above comments. We hope the comments above and changes made to the manuscript/SI are sufficient to be accepted for publication from Nature Communications but we are happy to make further refinements.

Reviewer #4 (Remarks to the Author):

The paper considers the application of artificial evolutionary techniques to droplet protocells and in particular considers the effects of changing the environment in which the evolution take place and whether interesting properties of the resulting phenotypes can be observed.

The novelty does not appear to lay in the evolutionary processes described in the paper, this aspects seems to be fairly standard genetic algorithm techniques and processes commonly used in many artificial population-based evolutionary set ups. The application to droplet protocells is however novel as fair as this review is aware.

The paper states that “the impact of environmental changes on an artificially evolving system has never been studied outside of computer simulations” – I believe that “never” in this context is too strong. Early work by Thompson, for example, considered evolution on electronic hardware (not simulation) and did consider environmental changes (such as temperature and radiation), work in the early to mid 90’s by JPL considered significant changes in environmental conditions with a for of field programmable transistor array (similar work was also conducted around the same time at the University of Heidelberg. More recently work related to the EU project SYMBRION consider evolution on real robots in the presence of changing environments. While I am not saying this is not an interesting experiment to carry out, I do not believe “never” is correct in this context.

We have changed the abstract and added the word “chemical”, to specify that the impact of the environment has never been studied before in evolving chemical systems. We do agree with this reviewer that the impact of the environment has been studied before in relation with evolutionary algorithms, but we think it is important to remark that we are referring here to evolution as in the context of biology (e.g. “open-ended evolution”), and not as a synonym to optimization, and how this can be applied to a chemical system.

Our system is very different to the SYMBRION project, for example, because there they have a finite set of programmed robots. The robots themselves are immutable, and it is their microcontroller what can be programed: based on an algorithm they reprogram the microcontroller following a genetic algorithm, for example. In our case the droplets are simple chemical entities. They are very mutable because we destroy them, and recreate again from scratch at the start of every experiment. Our algorithm can select their composition, but whatever they do depends on themselves and the different physical and chemical gradients. While in projects like SYMBRION the evolution happens purely *in silico*, in our system part of it happens *in silico*, and part of it happens *in vitro*. However, once again, we do appreciate the similarities and think that the juxtaposition of our work with the SYMBRION work is very important and interesting.

One key aspect is that to the best of our knowledge, no other study has studied the impact of the environment in real chemical entities, and the most similar one is the research performed by Lenski *et al.* where they changed the substrate where *E.Coli* was growing to citrate and showed that *E.Coli* could evolve to be able to grow in such conditions. Although their results were sensational, *E.Coli* is already a very advanced organism, while we focused on an extremely simple lipid aggregation, such as the ones that were available during abiogenesis, and we aimed to demonstrate how the environment could aid these very simple entities to evolve.

The paper states that “The objective was to create droplets that would provide a chemical compartment related to the aforementioned [which included: different polarities, densities, viscosities, solubility and interactions] attributes, aiming for their stability and the ability to move and divide”. The actual fitness used to drive evolution in these experiments appears to have been

“to maximise the number of active droplets after one minute of observation time”. It is not at all clear to me how this objective function (fitness function) can be used to measure let alone guide evolution to achieve these stated objectives?

We are sorry if this was not communicated properly. When we said “the objective was to create droplets that would provide a chemical compartment...” we meant that the reason why our chemistry set was chosen (1-Pentanol, 1-Octanol, DEP and Octanoic Acid) was with the objective of covering a different range of chemical properties, such as density or solubility. This is completely unrelated to the fitness function used within the Genetic Algorithm. We have rephrased the part where the chemistry is introduced in the main manuscript in order to make this clearer.

A number of “surprising and unexpected behaviours” were observed and listed in the paper; I am not an expert in droplet behaviours so could not really comment on whether these behaviours were surprising or not, but I do find it difficult to see the relevance of these to the artificial evolution that is the focus of the paper?

We have removed the word “surprising”. We consider that it is important to explain how the different obstacles interacted with the droplets, because this had a direct impact on how by changing the environments (the obstacles) the composition of the droplets varied through the genetic algorithm.

Our first hypothesis was that by adding obstacles, the number of collision between droplets, and droplets and environment, would increase, which would also increase the number of times the droplets divided. What was surprising and /or unexpected to us was that the environment had actually a negative effect, as we explained on that paragraph.

We think that this expansion of the description of the system helps understand the dynamic and versatile nature of the chemical system even if not directly related to the main point of the paper it helps understand the scientific systemic (difficult) state that we had to work with.

It would have been interesting to have more analysis on how the artificial evolution change between having a stable environment and a changing environment? Were the individuals evolved during the changing environment experiments in some way more adaptive to environments they had not been evolved in than the stable individuals? Did they divide more? Did any of the attributes mentioned before change?

Please refer to SI-13, “environments tested against best genomes” in the SI. In this case, we took the best genomes from each environment, and tested them in each of the other two environments. There it can be seen that the droplets evolved in the empty environment failed to adapt to the pillars or caves environment, while the ones evolved in the pillars environment were able to adapt in each of the other two environments, and the ones evolved in the caves environment were able to adapt to the empty environment. Thus, our data seems to indicate that droplets evolved in a changing environment were better at adapting to other environments, and that the pillars environment had the highest selection pressure. Although the manuscript mentions this concept (see the last paragraph just before the discussion) this conclusion was not included in the main manuscript, because even though we performed the experiments to obtain the data, we did not think the results obtained were convincing enough to be included in the main manuscript.

Performing these experiments was very time consuming, and we deemed that we had enough data to answer the main hypothesis of this research (“does the environment have an impact on the evolution of our droplet system?”), and we decided to concentrate on that, instead of pursuing another questions that arose during this research, which we plan to pursue in follow-up research.

Environment is being used as an evolutionary pressure on the process. One would expect a GA (or other evolutionary mechanism) to produce different individuals when faced with a changed environment – that is what evolution does. Not really sure this is an unexpected result. What happens if you put a good individual from the L-system back into the empty arena, does it produce a good individual quicker now than when it evolved from scratch – has this exposure to various environments made individuals more adaptive?

Part of this was answered just before, and the data is contained in the SI (see SI-13).

Regarding the results being unexpected or not, we agree with the referee that it is expected that the environment will have an impact on evolution, animal science nowadays is an example of this. What we think was unexpected and interesting is to certify that the environment had an impact on the evolution of extremely simple chemical entities, formed only by fatty acids and oils, that themselves cannot evolve. Such simple entities were available during abiogenesis, and we think it is important to show how basic droplet protocells formed by simple lipid aggregates were subject to environmental pressure. The concept of chemical evolution in the context of abiogenesis has already been theoretically investigated, and basic research in this regard has been performed showing how for example density can be a selector in chemical evolution. In our case we are considering more complex behaviours, such as movement or division, and we are pairing it with changing environments, and we showed that all these variables had an impact on the evolution of the simplest lipid aggregates.

Finally, the experiment the reviewer describes, such as putting a good individual from the L-system back into the empty arena, is something that we have discussed before, and that we plan to test in the future. But as said, these experiments are very time consuming, and we wish to focus at this point on presenting our work in this communication but an extensive further we be done to follow up.

The paper states “This research also highlights that engineering the environment itself as a variable is a promising approach for the optimization of complex systems.” It would be interesting if this statement was expanded somewhat and examples given as to what the aim of doing this might be and where it could be applicable.

By explicitly introducing and varying the environment we hope to emphasise its role and importance in the expression of our system. The statement highlights that the environment in which the system of interest 'lives' should be considered more often when optimising specific properties of the system.

In many aspects, this concept is similar to the one of morphological computation and embodiment [1], where re-framing a problem is often a simple yet robust way to resolve challenging problems.

These concepts have been explored in robotics, for example with the recent development of soft robots that allow to grasp very complex objects seamlessly. Previously a lot of research was done on manipulation and grasping because robotic hands were made of rigid metals introducing really complex grasping dynamics, were the control system had to adapt to the shape of the object. By re-framing the problem and adding soft materials on the gripper, the control problem became much easier. See [2] for a good example.

[1] Pfeifer, Rolf, and Josh Bongard. How the body shapes the way we think: a new view of intelligence. MIT press, 2006.

[2] Brown, Eric, et al. "Universal robotic gripper based on the jamming of granular material." Proceedings of the National Academy of Sciences 107.44 (2010): 18809-18814.

We have added the first reference explained here into the main manuscript.

I do not think the title of the paper properly describes what the essence of the paper is about – “Chemical Robot” are there really robots in these experiments, chemical or otherwise? Surely robots are entities that perform a series of “complex” actions, I don’t really see any actions going on here.

When we spoke about a “chemical robot” we were talking about the automated platform, not about the droplets. In order to address these concerns we have changed (including the title) the words “chemical robot” by “3D-Printed Fluidic Chemorobotic Platform”.

Why does the evolution use number of active droplets as its fitness function? And why is a one minutes observation window appropriate?

We decided to use 1 minute as the observation time for several reasons. First of all, it is a rounded unit of time. Secondly, it is the same amount of time we used in our previous research, thus by using the same observation time, we were able to compare results and validate the new platform. Finally, these experiments are very time consuming (preparation, cleaning, experimenting) and we wanted to maximize the quantity of data obtained, thus we chose the minimum possible time we deemed was good enough to produce meaningful data.

Newer research being developed at the moment within our group is using different time scales, and it is producing very similar data. In any case, the observation window chosen will only have an impact on the compositional genome, and we consider that the overall significance of the results would be the same, whether it is a minute or any other observation window.

Regarding to the fitness function used, we decided to consider the number of active droplets at the end of the experiment because this fitness function is a convolution of the two fitness functions we used in previous research (movement, and division). Thus, by using a very similar function we were able to compare the results of the new platform with the old one, and validate them. It is also worth considering, as we explained in previous research, the division and movement are very common behaviours among many living entities, and that is the reason why they were chosen initially.

These points have now been addressed in the main manuscript.

Having a GA run 10 generations and a population of 20 individuals is quite small, I wonder if larger populations and more generations were ever used to see if the evolution had in fact stopped?

We do agree with this reviewer that 10 generations and 20 individuals is quite small. These values were chosen considering time constraint as the main priority. Each recipe (or individual) was tested five times, and this in total took 30 minutes. Thus testing 20 individuals took 10 hours, which is roughly a working day, and therefore 10 generations would take roughly two weeks. However, we would also like to emphasise that our evolutionary experiments are not simple optimization experiment using GA. Thus, we do not think it is necessary for our experiment to be repeated until it saturates.

Despite this very low population and generation size, we were able to show how the system can evolve, and how the environment can impact this evolution. Our objective was to show precisely this, and we think we have managed to do it with these values. Increasing the generation size, or the total number of generations, would probably increase the quality of the data, but it would answer the main hypothesis of this research in the same way.

I actually like the experiments outlined in this paper and think it could well be of interest to many researchers in a number of different fields. But for me at the moment it raises too many unanswered questions.

We would like to thank the reviewer for the positive comment. We hope our revised manuscript has clarified all the unanswered questions and that the key findings should be published.

Reviewers' comments:

Reviewer #1 (Remarks to the Author):

Both the changes introduced in the new version of the manuscript and the answers provided to the reviewers have clarified some of the issues/concerns I had in relation to this work.

However, I am afraid that the revised manuscript is still far from ready to be published in Nat Comm. Here below I explain some of the new (or renewed) criticisms. In addition, other various minor points can be found in the annotated pdf, attached.

1. Abstract: although the overstatements about the nature/originality of the work presented here have somehow been softened in the main text, there is a sentence in the abstract that has been kept (roughly) the same and it definitely sounds too strong a claim: 'the impact of environmental changes on an artificial chemical evolving system has never been studied outside of computer simulations.' If more stress is given to the term 'chemical'... then one should not forget that Sol Spiegelman inaugurated the field of 'artificial in vitro evolution' 50 years ago... And taking into account precisely that well-established field, I think it is not correct to state (in such a general way) that the effects of the environment on artificial evolutionary processes have not been explored (outside of computer simulations), even if the approach undertaken here is, indeed, very different.

2. Introduction: The first paragraph of the introduction overlaps excessively (in content and references) with the passage at the end of p. 3 and beginning of p. 4. One or the other should be erased or substituted with new text.

3. My suggestion is to erase the first paragraph of the current introduction and write something completely new about the traditional way of approaching artificial 'chemical evolution': namely, Spiegelman's approach (or the more contemporary SELEX methodologies), highlighting the novelty of the current approximation to the problem, and not just comparing the type of physical molecules and 'phenotypes' involved, or the 'genotypes', but also in relation to the way of implementing the artificial selection algorithm and, more importantly, with regard to the possibility of modifying the environment now as the evolutionary experiment is being run. In other words, use the new paragraph for the actual aims of this paper and, at the same time, avoid ignoring such an important line of research in the context of the molecular and evolutionary origins of life (the classical conception of the process of abiogenesis by chemists).

4. Figures 1 and 3 also involve quite a bit of overlap. I would fuse them into one. In addition, expanding the number of main Figures from 4 to 7 without a clear demand from the reviewers... (except for the recommendation to improve labels and legends) perhaps is going too far.

5. The numbering of the references should be properly checked before resubmission. Authors refer to their previous Nat Comm paper now with three different numbers in the main text (10, 19 and 17) if I am not mistaken. Please, pay more attention to these important details (see also corrections in the pdf attached).

6. Regarding the references, I am not convinced that the current ref. 7 (Shirt-Ediss et al. 2015) responds to what the text implies at that point. A very recent publication by the same group (Piedrafita et al. 2017, Sci. Rep.) seems much more appropriate.

7. Last but not least, I still miss a somewhat deeper explanation of the underlying behavior of the droplets. The authors respond to my original criticism (and to a very similar point made by rev. 3) by

saying that it is not the main objective of this contribution, and that the types of behavior the droplets are showing have already been reported in the specialized literature. This excuse is not at all satisfactory. A better account of how do the individual behaviors of the droplets connect with their population dynamics and the global evolutionary trends would constitute an enormous gain for the paper and for its impact on the scientific community.

Reviewer #4 (Remarks to the Author):

I am happy with the changes the authors have made following my comments and now feel the paper is acceptable for publication.

We thank the reviewer #1 for valuable and thorough comments. We incorporated them in the paper and addressed them point by point in the following text.

The referee comments are in bold, and our replies in normal type.

Reviewer #1 (Remarks to the Author):

1. Abstract: although the overstatements about the nature/originality of the work presented here have somehow been softened in the main text, there is a sentence in the abstract that has been kept (roughly) the same and it definitely sounds too strong a claim: ‘the impact of environmental changes on an artificial chemical evolving system has never been studied outside of computer simulations.’ If more stress is given to the term ‘chemical’... then one should not forget that Sol Spiegelman inaugurated the field of ‘artificial in vitro evolution’ 50 years ago... And taking into account precisely that well-established field, I think it is not correct to state (in such a general way) that the effects of the environment on artificial evolutionary processes have not been explored (outside of computer simulations), even if the approach undertaken here is, indeed, very different.

We don’t entirely agree – the exploration of environmental changes on artificial evolution has not experimentally been done before – the test tubes etc were not changed. The field of artificial evolution relates to selection processes done by humans on RNA. However, we have changed the paragraph to say: "Although evolution has been widely studied in a variety of fields from biology to computer science, still little is known about the impact of environmental changes on an artificial chemical evolving system outside of computer simulations."

2. Introduction: The first paragraph of the introduction overlaps excessively (in content and references) with the passage at the end of p. 3 and beginning of p. 4. One or the other should be erased or substituted with new text.

We agree, so we have deleted the passage from the end of p. 3.

3. My suggestion is to erase the first paragraph of the current introduction and write something completely new about the traditional way of approaching artificial ‘chemical evolution’: namely, Spiegelman’s approach (or the more contemporary SELEX methodologies), highlighting the novelty of the current approximation to the problem, and not just comparing the type of physical molecules and ‘phenotypes’ involved, or the ‘genotypes’, but also in relation to the way of implementing the artificial selection algorithm and, more importantly, with regard to the possibility of modifying the environment now as the evolutionary experiment is being run. In other words, use the new paragraph for the actual aims of this paper and, at the same time, avoid ignoring such an important line of research in the context of the molecular and evolutionary origins of life (the classical conception of the process of abiogenesis by chemists).

Thank you for this useful suggestion. We have now inserted a passage about the work by Spiegelman in the third paragraph (p.2 middle). We also mentioned a related work on in vitro evolution of Qb replicase-encoding RNA in water-in-oil emulsion, as an example of RNA evolution under modified environment (i.e. reactions in water-in-emulsions, in contrast to reaction in a bulk test tube in the original experiment).

4. Figures 1 and 3 also involve quite a bit of overlap. I would fuse them into one. In addition, expanding the number of main Figures from 4 to 7 without a clear demand from the reviewers... (except for the recommendation to improve labels and legends) perhaps is going too far.

The original four figures were split into seven figures in response to one of the comments from Reviewer #2. This was to improve the readability of each figures as some of the figures contained many figures. Moreover, Nature Communications allows up to 10 figures per article, thus we are still under this limit. We think that the current format is the best one considering that the distribution will be on-line and it is important that our work is understood by readers in many disciplines.

5. The numbering of the references should be properly checked before resubmission. Authors refer to their previous Nat Comm paper now with three different numbers in the main text (10, 19 and 17) if I am not mistaken. Please, pay more attention to these important details (see also corrections in the pdf attached).

Apologies, we have addressed this issue.

6. Regarding the references, I am not convinced that the current ref. 7 (Shirt-Ediss et al. 2015) responds to what the text implies at that point. A very recent publication by the same group (Piedrafita et al. 2017, Sci. Rep.) seems much more appropriate.

We have added this new reference as suggested.

7. Last but not least, I still miss a somewhat deeper explanation of the underlying behavior of the droplets. The authors respond to my original criticism (and to a very similar point made by rev. 3) by saying that it is not the main objective of this contribution, and that the types of behavior the droplets are showing have already been reported in the specialized literature. This excuse is not at all satisfactory. A better account of how do the individual behaviors of the droplets connect with their population dynamics and the global evolutionary trends would constitute an enormous gain for the paper and for its impact on the scientific community.

We have expanded paragraph 4 addressing this point, and providing more information about the underlying behaviour of our system.